# Unexpected PD-L1 immune evasion mechanism in TNBC, ovarian, and other solid tumors by DR5 agonist antibodies

Tanmoy Mondal[1,2], Gururaj N Shivange[1,2], Rachisan GT Tihagam[2], Evan Lyerly[1,2,3], Michael Battista[1,2,3], Divpriya Talwar[1,2,3], Roxanna Mosavian[1,2,3], Karol Urbanek[1,2], Narmeen S Rashid[4], J Chuck Harrell[4], Paula D Bos[4], Edward B Stelow[5], M Sharon Stack[6], Sanchita Bhatnagar[2,7,*] & Jogender Tushir-Singh[1,2,7,8,**] (iD)

## Abstract

**Lack of effective immune infiltration represents a significant barrier to immunotherapy in solid tumors. Thus, solid tumor-enriched death receptor-5 (DR5) activating antibodies, which generates tumor debulking by extrinsic apoptotic cytotoxicity, remains a crucial alternate therapeutic strategy. Over past few decades, many DR5 antibodies moved to clinical trials after successfully controlling tumors in immunodeficient tumor xenografts. However, DR5 antibodies failed to significantly improve survival in phase-II trials, leading in efforts to generate second generation of DR5 agonists to supersize apoptotic cytotoxicity in tumors. Here we have discovered that clinical DR5 antibodies activate an unexpected immunosuppressive PD-L1 stabilization pathway, which potentially had contributed to their limited success in clinics. The DR5 agonist stimulated caspase-8 signaling not only activates ROCK1 but also undermines proteasome function, both of which contributes to increased PD-L1 stability on tumor cell surface. Targeting DR5-ROCK1-PD-L1 axis markedly increases immune effector T-cell function, promotes tumor regression, and improves overall survival in animal models. These insights have identified a potential clinically viable combinatorial strategy to revive solid cancer immunotherapy using death receptor agonism.**

**Keywords** DR5; dual-specificity antibodies; immune evasion; PD-L1; solid tumors
**Subject Categories** Cancer; Immunology

## Introduction

T-cell-activating monoclonal and bispecific antibodies have shown great potential to enhance immunotherapy against tumors and have led to collective decline in overall death rates from cancer (Mellman *et al*, 2011; Brinkmann & Kontermann, 2017). Unfortunately, in case of solid tumors, the average clinical response of cancer immunotherapy and chimeric antigen receptor T (CAR-T) cell-based strategies is significantly lower as compared to leukemia and melanoma (Melero *et al*, 2014). Other than factors contributing to cellular and genetic tumor heterogeneity, immune exhaustion and limited infiltration of immune effector cells in solid tumor microenvironment (called immune "cold" tumors) remain the bottleneck for observed lower immunotherapy responses (Melero *et al*, 2014; Anderson *et al*, 2017). In support, various immune checkpoint targeting therapies work effectively against "immune hot tumors" that are adequately infiltrated with effector T cells (Haanen, 2017). Furthermore, surgical tumor debulking (Guisier *et al*, 2019) and combinatorial approaches such as cytotoxic drugs (and neoadjuvant) also work via effective breakdown of solid tumor mass (Opzoomer *et al*, 2019), which orchestrates immunogenic tumor microenvironment (Obeid *et al*, 2007) and enhances increased T-cell infiltration to improve anti-tumor response.

Unlike neoadjuvant and chemotherapy drugs, extrinsic apoptotic pathways instigate tumor breakdown and clearance without the associated toxicity (Narayan & Vaughn, 2015). Therefore, pro-apoptotic receptor agonist (PARA) therapy using Trail ligand (Apo2L) or epithelial cancer-enriched death receptor-5 (DR5/TRAIL-R2) activating antibodies gained significant attention in early twentieth

1 Laboratory of Novel Biologics, University of Virginia, Charlottesville, VA, USA
2 Department of Biochemistry and Molecular Genetics, University of Virginia, Charlottesville, VA, USA
3 Undergraduate Research Program, University of Virginia, Charlottesville, VA, USA
4 Department of Pathology, Massey Cancer Center, VCU, Richmond, VA, USA
5 Department of Pathology, University of Virginia, Charlottesville, VA, USA
6 Harper Cancer Research Institute, University of Notre Dame, Notre Dame, IN, USA
7 University of Virginia Cancer Center and Medical School, Charlottesville, VA, USA
8 DoD Ovarian Cancer Academy Early Career Investigator, Charlottesville, VA, USA
*Corresponding author. Tel: +1 434 982 6441; E-mail: sb5fk@virginia.edu
**Corresponding author (lead contact). Tel: +1 434 98 20465; E-mail: jogi@virginia.edu

century. PARAs activate extrinsic apoptotic pathway by oligomerizing DR5, a hallmark of TNF receptor superfamily members (Ashkenazi, 2015). Importantly, PARAs preferably activate extrinsic cell death in p53 mutant cancer cells (Wu *et al*, 1997; Ashkenazi & Herbst, 2008). As >75% of solid tumors carry p53 loss-of-function mutations, multiple DR5 agonist antibodies: lexatumumab (Marini, 2006), apomab (Camidge, 2008), AMG655 (Graves *et al*, 2014), and tigatuzumab (Forero-Torres *et al*, 2015), have been tested clinically after proven effective in various immunodeficient xenograft solid tumor models (Camidge, 2008; Kaplan-Lefko *et al*, 2010; Tamada *et al*, 2015). Recent efforts are also directed to generate and test the second generation of DR5 activating approaches (Tamada *et al*, 2015; Wajant, 2019). Other reports have also described Apo2L ligand-agonist antibody co-targeting and bispecific antibody-based approaches to increase anti-tumor DR5 signaling (Graves *et al*, 2014; Shivange *et al*, 2018; Wajant, 2019).

Sadly, all clinically tested DR5 agonist antibodies so far have failed to improve survival in phase-II trials even when given in combination of nanoparticle albumin-bound paclitaxel (nab-paclitaxel) neoadjuvant therapy against high DR5 expressing TNBC patients (Forero-Torres *et al*, 2015). On the contrary, nab-paclitaxel therapy has significantly improved the survival of metastatic TNBC patients if given in combination of anti-programmed death ligand-1 (PD-L1) immunotherapy and was recently approved by FDA (Aktas *et al*, 2019). Nab-paclitaxel + anti-PD-L1 combinatorial immunotherapy orchestrates both immune-independent and immune-dependent anti-tumor responses, respectively (Pardoll, 2012), while nab-paclitaxel + anti-DR5 therapy lacks immune activating component. Given that TNBC, ovarian, and other solid tumors carry elevated PD-L1 levels and considering the lack of immune activating function in combinatorial nab-paclitaxel + DR5 agonist therapy, here we sought to test the hypothesis whether PD-L1-mediated immune evasion potentially contributes to lower anti-tumor response of DR5 agonist antibodies. Using various clinical DR5 agonist antibodies, multiple tumor cell lines, and immune-sufficient tumor models, here we demonstrate an unexpected PD-L1 cellular and surface stabilization mechanism which is regulated by DR5 agonist-activated Rho-associated kinase-1 (ROCK1) and proteasome function downstream of death-inducing signaling complex (DISC).

# Results

## PD-L1 stabilization by DR5 agonist antibodies in solid tumors

In last decade, multiple ligand–receptor interactions on immune cells–tumor cells (*vice versa*) called immune checkpoints (ICPs) have accounted for T-cell dysfunction/exhaustion in various tumor types (Pardoll, 2012; Vonderheide, 2015). Humanized antibodies blocking these ICP interactions are highly compelling in generating curative immune response in clinical trials (Mellman *et al*, 2011; Pardoll, 2012). One key ICP ligand called PD-L1, present on many solid tumor cell types, inhibits activity of antigen-specific $CD8^+$ T cell via interacting with PD-1. This results in evasion and immune dysfunction in solid tumors. Given the elevated function of PD-L1 in solid tumors, we tested its expression in response to DR5 signaling upon agonist antibodies treatment (see Appendix Tables S1 and S3 for list of agonist antibodies used and their sequences). Unexpectedly, we observed PD-L1 stabilization in cellular lysates by various clinical DR5 agonists in tested solid cancer cell lines (Figs 1A–E, and EV1A and B, Appendix Fig S2A, see also Appendix Table S2 for list of cell lines used). Blockade of caspase-8 containing death-inducing signaling complex (DISC) and downstream of DR5 inhibited PD-L1 stabilization (Figs 1C and EV1A). Importantly, cellular lysate stabilized PD-L1 was recruited and increased on surface of various tumor cells as confirmed by surface biotinylating assays (Figs 1F and EV1B) and flow cytometry (Fig 1G and H and Appendix Fig S1A–C). We observed increase in both mean fluorescent intensity (MFI) of

---

**Figure 1.  DR5 agonist antibody surface stabilizes PD-L1 on solid cancer cells and tumors.**

A–C    Total PD-L1, PARP, cleaved caspase-3 from colon (Colo-205), lung (A549), pancreatic (PANK1), triple-negative breast cancer cell (MDA-MB-436), and ovarian cell (Cavo-3) lysates treated with indicated DR5 agonist antibodies named Lexa (lexatumumab), KMTR2, BaCa, and tigatuzumab. GAPDH is loading control.

D    Total PD-L1, CD47, Calreticulin, PARP from MDA-MB-436 cell lysates treated with indicated DR5 agonist antibodies ± caspase inhibitor Z-VAD. GAPDH is loading control. See also EV1A for data using additional cell line.

E    Total PD-L1 Western blotting signal from the lysates of OVCAR-3 (*n* = 3) and MDA-MB-436 (*n* = 3) cells after 6 h of treatment of 100 nM lexa (DR5 agonist antibody) was normalized to GAPDH. Representative blots are in Appendix S2A.

F    Surface biotinylation of PD-L1 from indicated tumor cells after indicated DR5 agonist treatments. Cleaved caspase-3 and PARP indicate activation of DR5 signaling. In lanes 3 and 4, KMTR2 was pre-neutralized either with recombinant DR5 (rDR5) or recombinant FOLR1 (rFOLR1). See also EV1B for additional cell line data.

G    Representative flow cytometry plots of PD-L1 from two different tumor cell lines treated with indicated DR5 antibodies. Secondary alone and IgG1 control are included (See Appendix S1 for additional plots).

H    Relative surface PD-L1 % cells, after DR5 agonist treatments (see Appendix Fig S1). Relative signal is normalized to surface PD-L1 in IgG1-treated cells in corresponding tumor cells and horizontal line in samples indicated mean (*n* = 3–4).

I    $0.5 \times 10^6$–$2 \times 10^6$ indicated tumor cells were injected subcutaneously in NOD.Cg-Prkdc[scid] Il2rg[tm1Wjl]/SzJ animals with Matrigel in PBS. When tumors appeared on animals (3–4 weeks), animals were i.p. injected with indicated DR5 agonists (4–6 doses), followed by tumor extraction and preparation of single-cell suspension isolation from tumors after indicated antibody treatments.

J    Relative surface PD-L1 % cells in indicated tumors after DR5 agonist treatments (see Appendix Fig S3). Relative signal is normalized to surface PD-L1 in IgG1-treated tumors and horizontal line in samples indicated mean (*n* = 2–6).

K    ER (−), PR (−), and HER2 (−) UCD52 patient-derived tumor tissue was xenografted in breast fat pad of NOD.Cg-Prkdc[scid] Il2rg[tm1Wjl]/SzJ mice, following by treatment with IgG1 or KMTR2 (50 μg, four doses). Harvested tumors were analyzed for PD-L1, CD47, and PARP in lysates.

L    KMTR2-treated (50 μg, four doses) UCD52 TNBC PDX tumors were stained for PD-L1 using immunohistochemistry (IHC). For additional images, see Appendix Fig S2C. Scale bar indicates 50 μm.

M    Total PD-L1 and CD47 blotting analysis from DR5-resistant MDA-MB-436 cell lines after indicated DR5 agonist treatment (50 nM, 6 h). For DR5-resistant cell generation and additional data, see Fig EV2A–C.

Data information: Error bars represent SD. In (E), (H), and (J), unpaired Welch's *t*-test was used to determine *P* values (*$P < 0.05$, **$P < 0.005$, ***$P < 0.0001$).
Source data are available online for this figure.

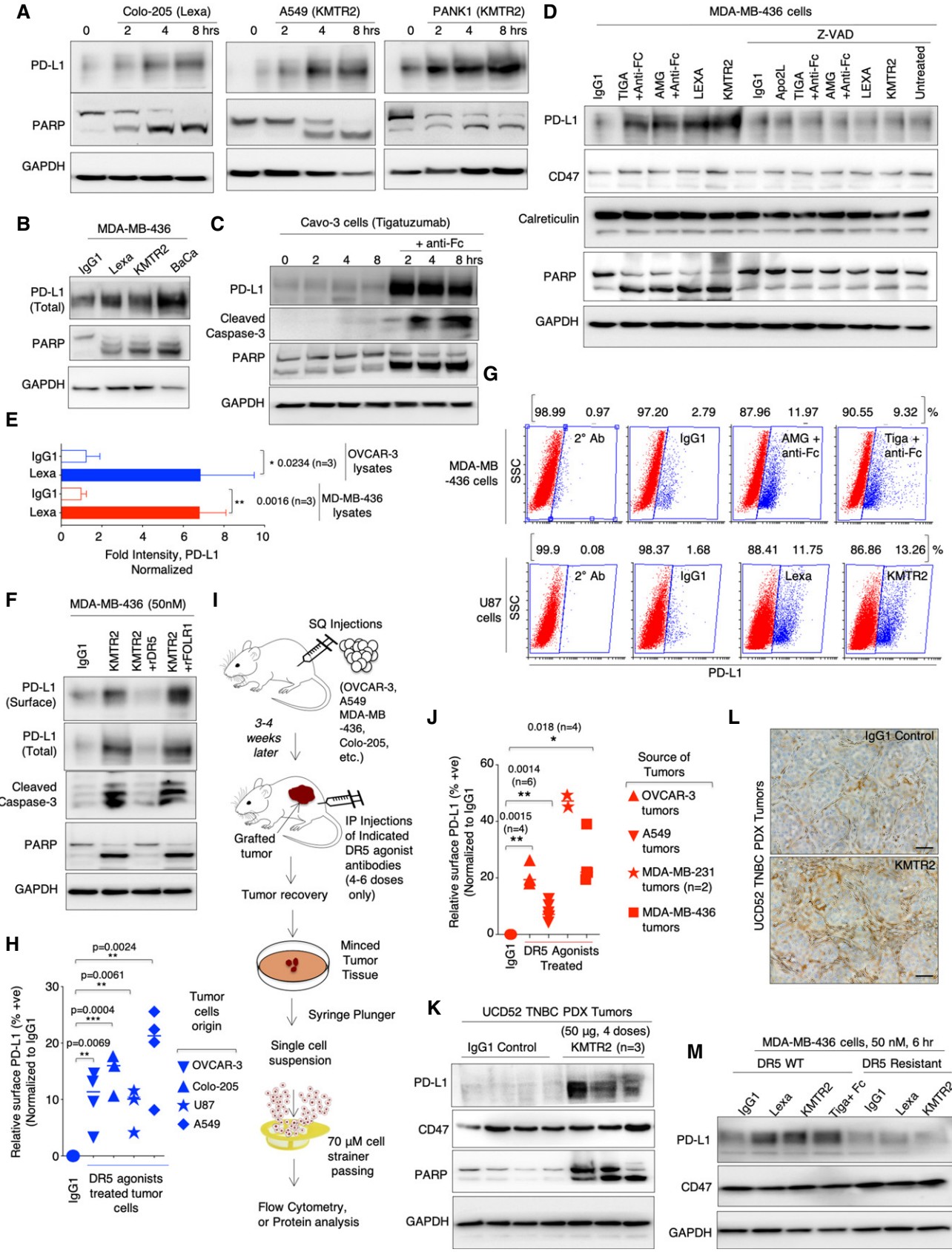

**Figure 1.**

PD-L1 and the relative % of cells expressing PD-L1 after various DR5 agonists treatment (Appendix Fig S2B). Next, we analyzed total and surface PD-L1 stabilization *in vivo* using cellular and patient-derived solid tumor xenografts treated with DR5 antibodies. All DR5 agonist antibodies were engineered with CH2 Fc L234A L235A (LALA) mutations to avoid interference from compliment system and NK cell-mediated antibody-dependent cellular cytotoxicity (Saunders, 2019). Isolated single-cell suspension of tumor cells (Fig 1I) from TNBC, ovarian, lung, and colon tumor cellular and TNBC PDX xenografts showed higher total and surface PD-L1 expression from animals treated with DR5 agonist antibodies as compared to IgG1 control (Fig 1J and K, Appendix Fig S3A–C). PD-L1 stabilization was also confirmed via IHC analysis using TNBC UCD52 PDX tumors treated with DR5 agonist, KMTR2 (Fig 1L and Appendix Fig S2C).

Given the inverse relation of tumor metastasis and patient survival, we next tested whether DR5 antibody-mediated PD-L1 stability would be lost during transient epithelial-to-mesenchymal transitions (EMTs). To this end, we induced transient EMT using A549 (and OVCAR-3 cells) as evident with loss of E-cadherin and gain of N-cadherin and vimentin (Fig EV1C). DR5 expression, sensitivity to DR5 agonists, and PD-L1 surface stabilization remained unaffected in these metastatic transformed cells (Fig EV1D–F). Next we generated DR5-resistant ovarian and TNBC cell lines via continuous exposure with the increasing concentration of lexatumumab for a period of 3 months (Fig EV2A and B). PD-L1 stabilization was not observed in ovarian and TNBC cells resistant to DR5 signaling (Figs 1M and EV2C).

## CSN5 is not required for PD-L1 upregulation by DR5 agonist antibodies

To test whether surface stabilized PD-L1 (by DR5 antibodies) will generate an immunosuppressive function, we made use of PD-1[+]

jurkat cells stably expressing luciferase reporter under NFAT response element (Fig 2A, Appendix Fig S4A–C). Treatment with anti-CD3 (OKT3) antibody induced luciferase activity in these reporter cells (Fig 2A–D). To confirm whether tumor cell surface mobilized PD-L1 will form complex with PD-1 checkpoint receptor, jurkat cells were co-cultured with DR5 agonist-treated tumor cells (Fig 2B–D, Appendix Fig S4B and C). DR5 agonist-treated ovarian, TNBC, lung, and colon cancer cell lines showed significantly reduced luciferase activity (Fig 2C, Appendix Fig S4C), confirming PD-L1-PD-1 interaction, a perquisite for T-cell dysfunction and immunosuppression. Strikingly, PD-L1 along with CMTM6 (Burr *et al*, 2017) a newly identified PD-L1 regulator were stable in tumor cell lysates for significantly longer hours (Appendix Fig S4D). When tested in DR5-treated tumor cell-jurkat cell co-culture reporter assays, inhibition of various posttranslational PD-L1 stability regulators (Hsu *et al*, 2018) such as mTOR, STAT3, CDK1, and NF-kβ did not change PD-L1 surface expression (Fig 2D). In addition, both transcription inhibitor (actinomycin-D) and translation inhibitor (cycloheximide) did not also reverse PD-L1 stability generated by DR5 agonists when compared with untreated or IgG1 control cells (Fig EV2D and E). Similar to cellular lysate results (Fig 1C), inhibition of DISC-caspase-8 activity reduced PD-L1 activity in reporter assays (Fig 2D). Taken together, these results indicated that PD-L1 stabilization (after DR5 agonist treatments) requires the function of a regulatory pathway, which potentially is not modulated by transcription or translation function rather by the DISC and caspase activation.

Since DR5 ligand Apo2L belongs to TNF superfamily and TNF-α was recently shown to stabilize PD-L1 by activating a deubiquitinase COP9 signalosome 5 (CSN5) enzyme (Lim *et al*, 2016), we tested whether DR5-mediated PD-L1 stabilization also requires CSN5. TNBC tumor cells were treated with DR5 agonist antibodies and TNF-α next to each other. TNF-α stabilized both PD-L1 and CSN5 in ovarian and TNBC tumor cells without activating caspases

---

**Figure 2. PD-L1 stabilization is CSN5 independent but protease dependent.**

A Model of CD3 activation that induces luciferase in PD-1 effector jurkat reporter cells. Also see Appendix Fig S4.

B Tumor cell-Jurkat cell co-culture model: Upon DR5 activation in tumor cells, surface mobilized PD-L1 engages PD-1 on jurkat reporter cells leading to loss of luciferase activity. Also see Appendix Fig S4.

C Luciferase activity of reporter lines from tumor cell-Jurkat cell co-culture assay using MDA-MB-436 and OVCAR3 cells after treatment (50 nM) with indicated DR5 agonists (tiga: tigatuzumab, AMG: AMG655, KMTR1, Lexa: Lexatumumab). The background luciferase signal from untreated cells (due to basal surface PD-L1) was subtracted. Various controls treatments (IgG1, anti-PD-L1, anti-EGFR) are also shown (*n* = 3).

D Same as (C) except tumor cells were pre-treated with indicated inhibitors for AKT, ERK, mTOR, MEK, STAT3, PARP-i, and NF-Kβ along with DR5 agonist KMTR2 antibody prior to co-culture.

E–G MDA-MB-436 cells were treated with TNFα and indicated DR5 agonists for indicated times. Lysates were analyzed for PD-L1, CSN5, phosphorylated p65, total p65, cleaved caspase-3, and PARP. GAPDH is loading control in (E and G). Tubulin is loading control in (F). Additional data from OVCAR-3 are shown in Fig EV2F.

H Flow cytometry analysis of MDA-MB-436 cells treated with TNFα ± MG132 (top) and indicated DR5 agonist ± MG132 (bottom).

I Total PD-L1 from OVCAR3 cell lysates was analyzed after treatment of indicated DR5 agonist ± MG132. Additional data from MDA-MB-436 are shown in Fig EV2G.

J Flow cytometry surface PD-L1 analysis of OVCAR-3 cells treated with indicated DR5 agonist ± MG132.

K MDA-MB-436 cells were treated with MG132 for indicated times, and total PD-L1 from lysates was analyzed. GAPDH is loading control.

L Immunoblotting of DR5 confirming generation of knock out (DR5-KO) cell lines.

M Cell viability analysis of DR5-KO and DR5-WT cells (*n* = 3).

N DR5-KO and WT cells were treated with TNFα and indicated DR5 agonist followed by flow cytometry using PD-L1 specific antibodies. See also Appendix Fig S5B.

O TNBC WT and DR5-KO (MDA-MB-436) cells were treated with either DR5 agonist or TNFα as indicated. Lysates were analyzed for PD-L1, S5a, and DR5. GAPDH is loading control.

P OVCAR-3 and MDA-MB-436 (WT and DR5-KO) cells were treated with KMTR2 followed by ubiquitin and S5a immunoblotting from total lysates. GAPDH is loading control. Blue arrow indicate decreased S5a levels and red brackets in corresponding lanes (below) shows overall increased ubiquitin signal in lysates.

Data information: Error bars in (C), (D), and (M) represent SD. Unpaired Welch's *t*-test was used to determine *P* values for (C) and (D). Paired *t*-test was used for (M) (**$P < 0.005$, ***$P < 0.0001$).
Source data are available online for this figure.

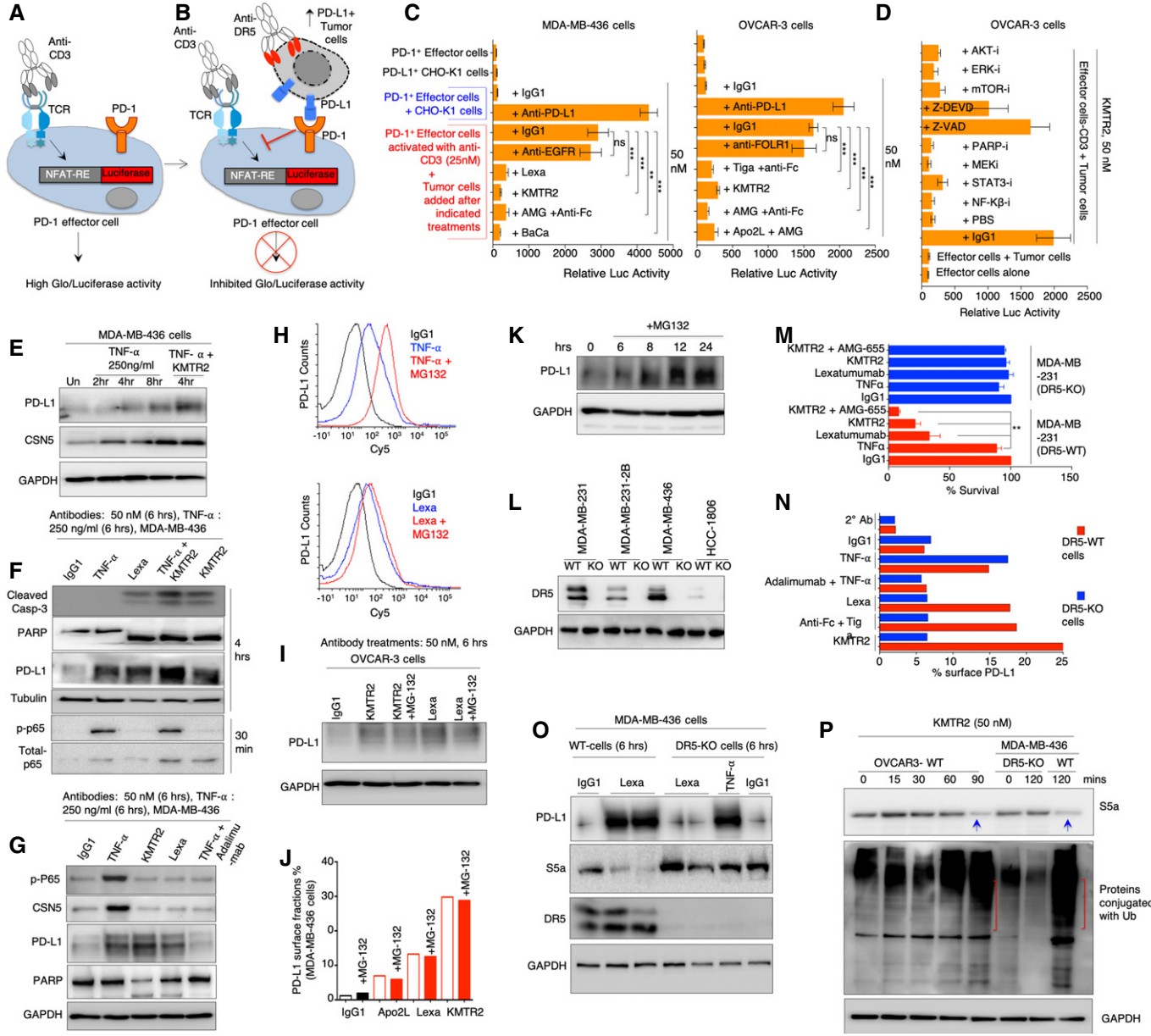

**Figure 2.**

(Figs 2E–G, and EV2F). CSN5 upregulation was not detected in DR5 agonist-treated lysates (Figs 2E–G, and EV2F). Mechanistically, CSN5 is a deubiquitinase and functions by removing ubiquitin tags from PD-L1. Thus, CSN5 inhibits PD-L1 degradation by proteasome complex (Lim *et al*, 2016). Similar to published report (Lim *et al*, 2016), we also observed increased PD-L1 basal stability after TNF-α and a proteasome inhibitor (MG132) co-treatment (Fig 2H top). Strikingly, proteasome inhibition (for a longer period) increased overall PD-L1 in tumor cells lysates (Fig 2K) and DR5 agonist plus MG132 co-treatment did not additionally stabilize PD-L1 on cell surface or in total lysates (Figs 2H–J, and EV2G, Appendix Fig S5A). These results indicated that proteasome inhibition (by MG132) potentially is a linear and downstream event of DR5 agonist signaling. As DR5 agonist antibodies function via direct caspase-8

activation in DISC and given the reports of caspase-mediated proteasome inactivation (Cohen, 2005), we next explored the possibility that PD-L1 stabilization is a byproduct of proteasome inactivation caspases.

To this end, we generated DR5 knockout (DR5-KO) TNBC cells using CRISPR-Cas-9 approach (Fig 2L). Generated DR5-KO lines were not sensitive to DR5 agonists (Fig 2M) and did not mobilize PD-L1 on tumor cell surface (Fig 2N, Appendix Fig S5B). It must be noted that TNF-α-mediated PD-L1 stabilization remained unchanged in DR5-KO lines as compared to DR5-WT cells (Fig 2M and N, Appendix Fig S5B). If DR5 agonist stabilize PD-L1 via proteasome inactivation, we hypothesized significantly higher degradation of proteasome regulatory submits in DR5-WT tumor cell lines as compared to DR5-KO cells upon DR5 agonist treatments. Since

various proteasome regulatory subunits of 26S proteasome (such as S6', S1, and S5a/PSMD4) are known to be cleaved during proteasome inactivation (Sun *et al*, 2004; Cohen, 2005), we next tested total S5a/PSMD4 levels after DR5 agonist treatments in WT and DR5-KO cells. As expected, the proteasome's regulatory submit S5a/PSMD4 was only degraded in MDA-MB-436 WT cells but not in DR5-KO cells (or DR5-resistant cells) after DR5 agonist treatment (Figs 2O and P, and EV2H). Furthermore, TNF-α signaling stabilized PD-L1 without affecting S5a levels (Fig 2O, lane 6 vs lanes 2, 3). Considering S5a degradation by DR5 agonists represents interference with proteasome function, overall ubiquitin signal increased only in DR5-sensitive cell lysates but not in DR5-KO cells after indicated antibody (Lexa or KMTR2) treatments (Fig 2P). Collectively these results confirm two different proteasome interference mechanisms of PD-L1 stabilization, where one works by deubiquitinating PD-L1 (Lim *et al*, 2016), while other works by degrading proteasome subunits.

## Role of ApoEVs in PD-L1 stabilization in heterogenous tumors

Our collective results (Figs 1 and 2) suggest that targeting DR5 agonists against homogeneously sensitive tumor cells (having cytotoxicity above apoptotic threshold) will generate superior anti-tumor cell death (Fig 3A). Aftereffect, with every single tumor cells eliminated [such as in DR5-sensitive homogenous cellular xenografts in immunodeficient mice (Motoki *et al*, 2005; Zhang *et al*, 2007; Camidge, 2008; Kaplan-Lefko *et al*, 2010)], surface PD-L1 will potentially have limited consequence even if PD-1 expressing T cells were present. Considering that strong PD-L1 stabilization was evident in UCD52 mixed TNBC PDX tumors (Fig 1K and L) and that all human tumors are heterogeneous and given that all tested DR5 agonists have failed in clinical trials, a scenario can be envisioned where surface stabilized PD-L1 on dying (highly DR5 sensitive) tumor cells influences the function of neighboring non-

dying cells (not DR5 sensitive) having cytotoxicity below apoptotic threshold (Fig 3A). In this scenario, stabilized PD-L1 has potential to suppress the activity of incoming cytotoxic T cells in tumor microenvironment (TME) and it will give survival advantage to DR5-resistant tumor cells (Fig 3A). To investigate above, we tested the hypothesis if surface PD-L1 from dying cells is shuttled to resistant and non-dying cells via apoptotic cell-derived extracellular vesicles called ApoEVs (Caruso & Poon, 2018; Gregory & Dransfield, 2018). We confirmed significantly high PD-L1 presence in ApoEVs after DR5 agonist treatment using dot blot and Western immunoblotting (Fig 3B–D). Next, we made use of ʟ-homopropargylglycine (HPG, a methionine analogue) incorporation instead of methionine in cultured cells (Calve *et al*, 2016). By making use of HPG specific catalyzing dye in flow cytometry studies, we confirmed PD-L1 transfer from WT cells to DR5-KO cells via ApoEVs (Fig 3E and F). To this end, we isolated ApoEVs from DR5-WT cells (grown in HPG+/Met- or Met+ media) after agonist antibody treatments. Next we incubated Met+ and HPG+ ApoEVs with DR5-KO cells for 48 h. PD-L1 signal with HPG selective dye in DR5-KO cells (grown in the absence of HPG) confirmed PD-L1 transfer from WT cell (grown in HPG+ culture media; Fig 2F). Next we tested ApoEVs mediated PD-L1 transfer kinetics using a time course experiment. We observed a significantly high surface and total PD-L1 levels after 24–48 h of incubation times (Fig 3G and H, Appendix Fig S6). As expected, direct DR5 antibody treatments did not stabilize PD-L1 on DR5-KO cells (Fig 3I). Furthermore, when tested next to each other, unlike DR5 agonist antibodies, TNF-α signaling which also stabilized PD-L1 did not shuttle significant amount of PD-L1 via ApoEVs (Fig EV3A and B). Importantly, DR5-KO cells incubated with ApoEVs showed significantly reduced luciferase activity when co-cultured with PD-1+ jurkat cells (Fig 3J) confirming PD-1-mediated T-cell inhibition by ApoEVs transferred PD-L1. Collectively, these results strongly support orchestration of DR5 agonist-mediated PD-L1 stabilization in heterogenous culture

**Figure 3. ApoEVs and non-apoptotic caspase-8 help stabilize PD-L1 in DR5 insensitive tumor cells.**

A   Schematic showing presence of tumor cells capable of optimal and non-optimal apoptotic activation by DR5 agonists constitute heterogeneous tumors.

B–D   ApoEVs isolated after IgG1 and KMTR2 treatment (OVCAR-3) were blotted against CD63 and PD-L1 in dot blots. ApoEVs isolated after IgG1 and tigatuzumab (C) and lexatumumab (D) treatment (MDA-MB-436) were run on western and blotted against CD63 and PD-L1.

E   Details of experimental data described in (F).

F   ApoEVs isolated from DR5-sensitive tumor cells grown in Met−HPG+ media were added on to DR5-KO cells (growing in regular media). After 48 h flow cytometry analysis was carried out with HPG catalyzing dye. HPG incorporation (3rd plot, see Methods for more details) from flow cytometry data confirms ApoEV transfer from cells growing in Met−HPG+ (DR5-WT) to DR5-KO cells.

G   ApoEVs isolated from DR5-WT OVCAR-3 cells (KMTR2 treated) were added on to DR5-KO (MDA-MB-231) cells for indicated times (24–96 h) to analyze PD-L1 transfer kinetics from ApoEVs (see also Appendix Fig S6).

H   Same as (G) except after addition of MDA-MB-436-derived ApoEVs (treated with combination of KMTR2 + lexatumumab DR5 agonists or IgG1) on to DR5-KO MDA-MB-231 cells, total lysates were immunoblotted for PD-L1 after 24 h. KMTR2 + lexatumumab (*n* = 3), IgG1 (*n* = 2).

I   Left: PD-L1 surface histogram shows direct antibody treatment on DR5-KO cells. Right: PD-L1 surface histogram from DR5-KO cells after treatment (24 h) with ApoEVs isolated from DR5-sensitive cells after either Lexa or KMTR2 treatment.

J   Same as (H and I) except ApoEVs treated tumor cells were analyzed in PD-1 reporter co-culture assays for luciferase signal (*n* = 3; see Fig 2).

K   MDA-MB-436 cells were treated with KMTR2 for 6 h and along with Z-DEVD for indicated times. Z-DEVD shows final time of KMTR2 exposure to cells in the absence inhibitor. Lysates were immunoblotted for cleaved caspase-3, 8, PARP, PD-L1, S5a, ROCK1 and GAPDH (*n* = 3). u.c: uncleaved, c: cleaved, n.s: non-specific band.

L   Relative caspase-8 and caspase-3 activity assays from (K). Similar to cleaved caspase-8 profile in (K), caspase-8 maintained steady activity, while caspase-3 activity required at least 40 (+) mins of DR5 agonist treatment without Z-DEVD (*n* = 3).

M   After indicated treatments with KMTR2 (DR5 agonist) and Z-DEVD (caspase-3 preferred inhibitor) similar to (K and L), cells were allowed to grow 24 h, followed by cell viability analysis.

Data information: Error bars in (J) represent SD. Error bars in (L) and (M) represent SEM. For statistical significance, *t*-test was used for (J), (L), and (M; **$P < 0.005$, ***$P < 0.0001$, ****$P < 0.00005$).

Source data are available online for this figure.

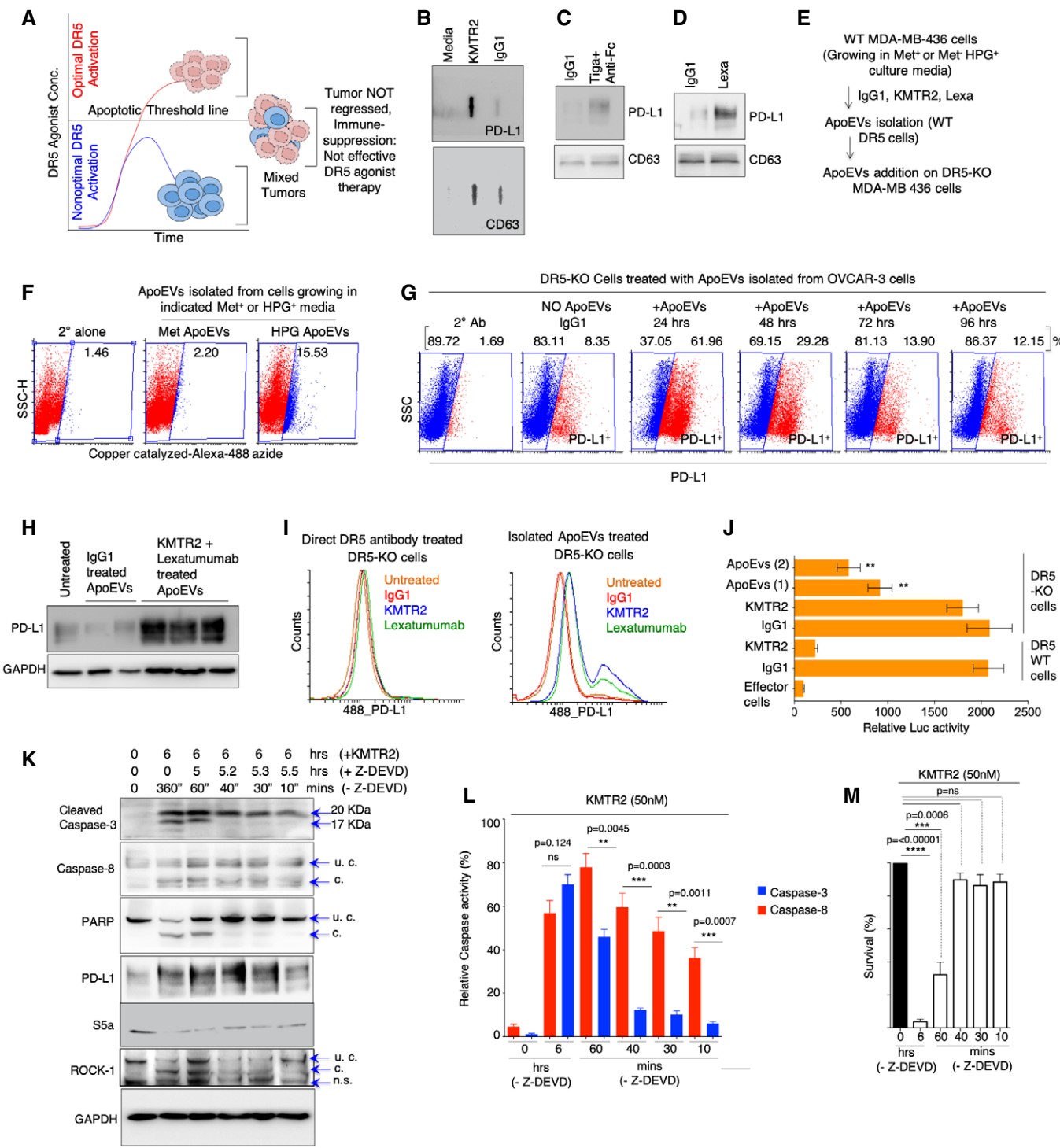

**Figure 3.**

conditions due to ApoEVs mediated DR5 transfer from extrinsic apoptotic sensitive cells to resistant cells.

As many clinically failed DR5 agonists activate apoptosis below the tumor clearance threshold (Ashkenazi, 2015), next we asked whether PD-L1 stability is maintained after DISC activation in the absence of complete execution of cell death. To this end, we established condition by making use of Z-DEVD, which preferentially

inhibits mainly autocatalytic function of caspase-3 required for its conversion from 20KDa form into fully activated 17KDa form (Ponder & Boise, 2019). With KMTR2 and Z-DEVD co-treatments, we observed steady activity of caspase-8 (as evident by its cleavage), while the activity of pro-domain containing cleaved caspase-3 was inhibited (Fig 3K), if Z-DEVD was added to the cultures within 40 min of DR5 agonist (KMTR2) treatment. The latter is evident by

loss of caspase-3 autocatalytic event (lack of accumulation of 17KDa, see lanes 4–6 in Fig 3K) and loss of PARP cleavage (see lanes 4–6 in Fig 3K). The results were also supported by caspase-3 activity assays (Fig 3L) and cell death assays (Fig 3M). Importantly, S5a levels were also reduced and PD-L1 was stabilized in lysates despite lack of effective cell death (Fig 3K, lanes 2–6). These findings are also suggestive that DR5 agonist antibodies with high extrinsic ability to generate activated DISC-caspase-8 (despite not executing cell death above tumor clearance threshold) can also potentially stabilize PD-L1. The latter could be a contributing factor in tumor cells having dysregulation of downstream pro-survival proteins such as Bcl-2 (and Bcl-xL etc.) despite optimal DR5 activation.

### DR5 agonist-activated ROCK1 is required for PD-L1 surface mobilization

A phase-II combination trial of paclitaxel and DR5 agonist tigatuzumab in TNBC patients has described upregulation of genes involved in apoptotic blebbing (Forero-Torres et al, 2015). Strikingly, activation (cleavage) of ROCK1, the key regulator of membrane blebbing, cytoskeletal ruffling, and vesicular exocytosis (Coleman et al, 2001), was maintained (Fig 3K) despite autocatalytic blockade of caspase-3, PARP and loss of cell death (Fig 3L and M), suggesting potential role of DISC-caspase-8 in activating ROCK1 downstream of DR5 agonists. Interestingly, a very recent study discovered ROCK1 (Rho-associated coiled coil containing kinase-1) to be PD-L1-associated regulator using ingenuity pathway analysis (Chan et al, 2019). Furthermore, ROCK1 is PD-L1 co-expressed gene in melanoma patients (Madore et al, 2016). Given that PD-L1 must be kept away from lysosome (Burr et al, 2017) and proteasome (Lim et al, 2016) for its stability, we next investigated whether DR5 agonists require ROCK1 function to surface mobilize PD-L1. We discovered early activation of ROCK1 and myosin light chain phosphorylation by all DR5 agonists in various tumor lines (Figs 4A–C, and EV3C–G). Similar to our observations with Z-DEVD inhibitor (Fig 3K), a partial ROCK1 cleavage was evident prior to full caspase-3 activation (Figs 4A and B, and EV3C and D, compare red arrows in lanes). These results indicated that in scenarios where caspase-3 function is inhibited, the upstream DISC-caspase-8 also have potential to activate ROCK1 pathway. Regardless, various ROCK1 inhibitors (ROCK1i) downregulated surface PD-L1 levels without effecting cell death (Fig 4D and E and Appendix Fig S7). In addition, during native immunoprecipitation (IP) studies with clinical anti-PD-L1 antibody (avelumab), activated ROCK1 directly interacted with surface PD-L1 in a complex that also contained CMTM6 (Fig 4F–H and Appendix Fig S8A–C). Interestingly, DR5 agonist + ROCK1 inhibition did not affect the stabilized PD-L1 levels in cellular lysates (Fig 4G, 5th lane, right blot), confirming PD-L1 intracellular stabilization being independent of ROCK1 activity but dependent on DISC-caspase-8 function. When DR5-treated tumor cells and jurkat co-culture reporter assays were evaluated in the presence of ROCK1 inhibitors, luciferase activity was significantly enhanced in lieu with loss of surface PD-L1 (Fig 4I and J and Appendix Fig S8D). Next, we tested ROCK1 inhibitor GSK269962 (2 mg/kg) in combination of murine DR5 agonist MD5-1 (50 μg dose) in 4T1 syngeneic TNBC tumor models. We observed significant tumor reduction by antibody drug combination over MD5-1, while GSK269962 alone was ineffective (Fig 4K). All together, these findings demonstrate that ROCK1 downstream of DR5 agonists plays an important role to help mobilize (potentially via CMTM6 or other unknown mechanisms) the internally stabilized PD-L1 on tumor cell surface.

---

**Figure 4. DR5 agonist-activated ROCK1 functions to help PD-L1 surface mobilization.**

A, B   KMTR2, KMTR2 + Z-VAD, and lexatumumab-treated MDA-MB-436 and OVCAR3 (respectively) lysates for indicated early time points were analyzed for caspase-8, caspase-3, ROCK1, pMLC, PARP, PD-L1, and CMTM6 as indicated. Red arrows indicate sequential kinetics of caspase-8, ROCK1, and caspase-3 activation, blue arrows indicate cleaved and activated caspase-8, ROCK1, and caspase-3, and black arrows indicates non-specific (n.s.) band by ROCK1 antibody.

C   OVCAR3 cells were treated with pMLC activating ionophore A23187 (positive control) and lexatumumab ± ROCK1 inhibitors or ± rDR5 or ± rFOLR1. Lysates were later analyzed for ROCK1, pMLC, caspase-3, and PARP with GAPDH as loading control. Blue arrows indicate cleaved and activated ROCK1 and caspase-3.

D   MDA-MB-436 and OVCAR3 cells were treated with indicated ROCK1 inhibitors 2 h prior to DR5 agonist KMTR2 treatment (20 nM). After 6 h flow cytometry was used to analyze surface PD-L1. (GSK269: GSK269962A, GSK429: GSK429286A).

E   MDA-MB-436 cell survival assay after treatment with DR5 agonist antibody KMTR2 ± ROCK1 inhibitors, error bars indicate SD (n = 2). (GSK269: GSK269962A, GSK429: GSK429286A).

F   Schematic of immunoprecipitation assay shown in (G and H). Cultured MDA-MB-436 cells were treated with KMTR2 (IgG1-Fc) or IgG1 control ± ROCK1i (GSK269962) for 6 h. After 6 h, 500 nM anti-PD-L1 avelumab (IgG4-Fc) was added to the media for additional 1 h. Cellular lysates were pulled down with anti-hu-IgG4 specific beads.

G   Immunoprecipitated lysates and leftover supernatant after various treatments as described in (F) were run at the same time followed by blotting with ROCK1, PD-L1, and CMTM6. Total lysates as a control were also generated in exactly similar conditions and run next to supernatant and IP samples. Blue arrows indicate cleaved and activated ROCK1.

H   Immunoprecipitation assays with anti-PD-L1 (avelumab) same as (G) using OVCAR-3 cell lines. Blue arrow indicates cleaved and activated ROCK1.

I   After treatment with IgG1 control and indicated DR5 agonists (KMTR2, Lexa, Tigatuzumab) ± ROCK1 inhibitors (GSK269962A, GSK429286A) tumor cells were co-cultured with anti-CD3 stimulated PD1+ effector Jurkat cells (stably expressing luciferase under NFAT-RE promoter). Relative luciferase signal was quantified and plotted after subtraction of background signal from untreated cells.

J   Relative luciferase activity from assay as described in (I). Data are from four different DR5 agonist antibodies used in combination of two different ROCK1i inhibitors (n = 3).

K   Tumor regression efficacy of MD5-1 (50 μg), GSK269962A (<2 mg/kg) and MD5-1 + GSK269962A (50 μg + 2 mg/kg) 4T1 tumors (n = 4–7). 6- to 8-week-old BALB/c mouse bearing 4T1 tumors (~100 mm³) were intraperitoneally (i.p.) injected with 50 μg of indicated antibody every third day (n = 4–6). GSK269962A (ROCK1i) was injected directly into the tumors. Tumor volumes were quantified at indicated days by caliper measurements.

Data information: Error bars in (J) represent SD. Statistical significance in (J) was determined by unpaired Student's t-test and in (K) using two-tailed paired Wilcoxon–Mann–Whitney test (*P < 0.05, **P < 0.005).
Source data are available online for this figure.

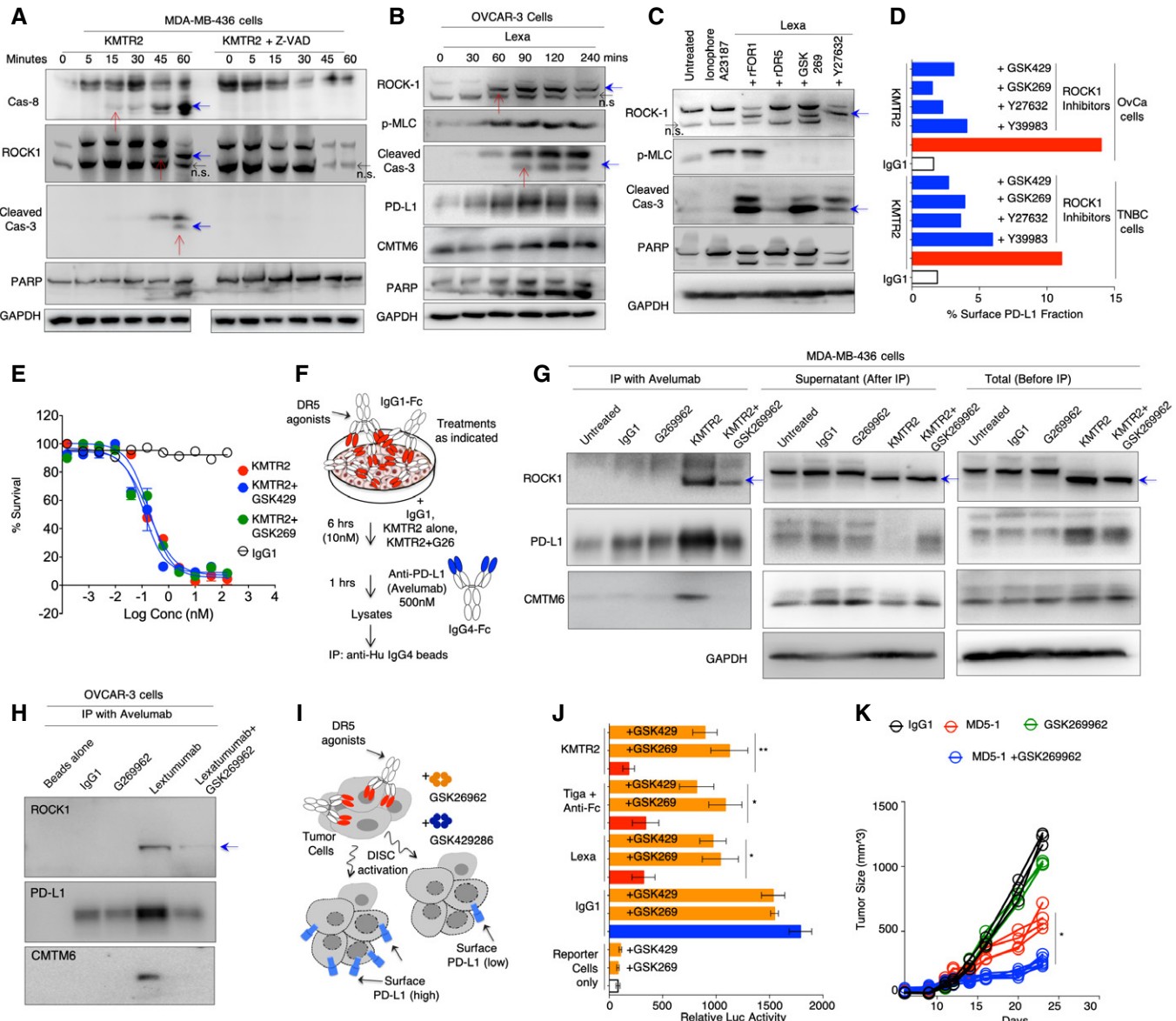

**Figure 4.**

## Generation of chimeric receptor to test clinical human DR5 agonists in immunocompetent mouse models

To investigate clinical human DR5 agonist using *in vivo* heterogeneous mixed tumor graft condition, we engineered chimeric DR5 constructs, with extracellular domain (ECD) of human DR5 fused with mouse DR5 transmembrane (TM) and intracellular domain (ICD; Fig 5A). These constructs were expressed in murine 4T1, MC38, and ID8 cells using lentiviral infection. We observed mixed stable lines with >85% cells being positive for DR5 (Fig EV4A). Interestingly, full-length human DR5 (huDR5) and G4S linker fused (chi-G4S-DR5) expressed in murine cells, while a direct fusion (chi-DR5) was not stable (Fig 5B–D, see Appendix Table S4 for sequences of chi-DR5 and chi-G4S-DR5). In terms of activity, consistent with its expression, only chi-G4S-DR5 was activated with clinical DR5 agonist antibodies to induce ~75–85% cell death (Fig 5E).

When grafted on BALB/c animals, only 4T1 chi-G4S-DR5 cells formed tumors, while huDR5 stable 4T1 tumors were rejected (Fig 5F). Lexatumumab-treated chi-G4S-DR5 mixed tumors had higher overall PD-L1 on cell surface, while co-treatment with clinical anti-PD-L1 antibody avelumab (human/mouse cross-reactive) inhibited surface presence of PD-L1 (Fig 5G and Appendix Fig S9). This is in agreement with published reports of glycosylated PD-L1 internalization after binding to activity blocking anti-PD-L1 antibodies (Lee *et al*, 2019).

## DR-5-ROCK1/DR5-PD-L1 co-targeting improves efficacy of clinical human DR5 agonists

Consistent with higher PD-L1 in chi-G4S-DR5 stable 4T1 tumors (Fig 5G), DR5 agonist and ROCK1 inhibitor co-treatment (similar to MD5-1-ROCK1 co-targeting) showed higher anti-tumor efficacy

(Fig 5H and I, tumors isolated after seven doses) as compared to lexatumumab alone. To confirm that surface enhanced PD-L1 in chi-G4S-DR5 stable 4T1 tumor works by inhibiting effector CD8$^+$ T-cell function via PD-1 engagement, next we depleted CD8$^+$ T cells in tumor-bearing animals (Fig 5J and K). *In vivo* depletion of CD8$^+$ T cells abrogated the combinatorial efficacy of lexatumumab and ROCK1 inhibitor (Fig 5L, tumors harvested after six doses). Similar higher anti-tumor efficacy results were obtained when lexatumumab (or KMTR2) was used either in combination of anti-PD-L1 avelumab antibody (Fig 5M–P) or a PD-1 inhibitor (Fig EV4C and D). As expected, depletion of CD8 cells in tumor-bearing animals also abrogated combinatorial lexatumumab and avelumab efficacy (Figs 5M and EV4B, tumors harvested after six doses). We also carried out animal survival studies with double and triple co-targeting of DR5-PD-L1-ROCK1 co-targeting. Consistently double and triple combinations significantly improved survival as compared to DR5 agonist alone or anti-PD-L1 alone (Fig 5Q).

To further confirm the direct role of intra-tumor effector CD8$^+$ T cell in improved combinatorial efficacy, animals were treated with four doses of DR5 agonist + ROCK1 inhibitor (and avelumab). Next, size-matched tumors were later subjected to tumor-infiltrating leukocyte (TIL) enrichment using Ficoll-Paque method as described previously (Tan & Lei, 2019) followed by dual staining T-cell analysis using flow cytometry. We observed significant enriched of CD8$^+$CD45$^+$ and CD4$^+$CD45$^+$ T cells in DR5 agonist-treated tumors regardless of ROCK1i and avelumab co-treatments (Fig 6A and B, Appendix S10, $n = 6–20$ tumor-bearing animals, three separate experiments). Similar results of immune infiltration were evident when representative chi-G4S-DR5 stable MC38 tumors treated with either DR5 agonist alone or in combination of ROCK1i or avelumab were tested in CD8, CD4 IHC studies (Fig EV5). Collectively, these observations support tumor breakdown (debulking) function+ of DR5 agonists to enhance immune infiltration. Based on numerous clinical trials, a higher ratio of cytotoxic T cells over helper T cells has

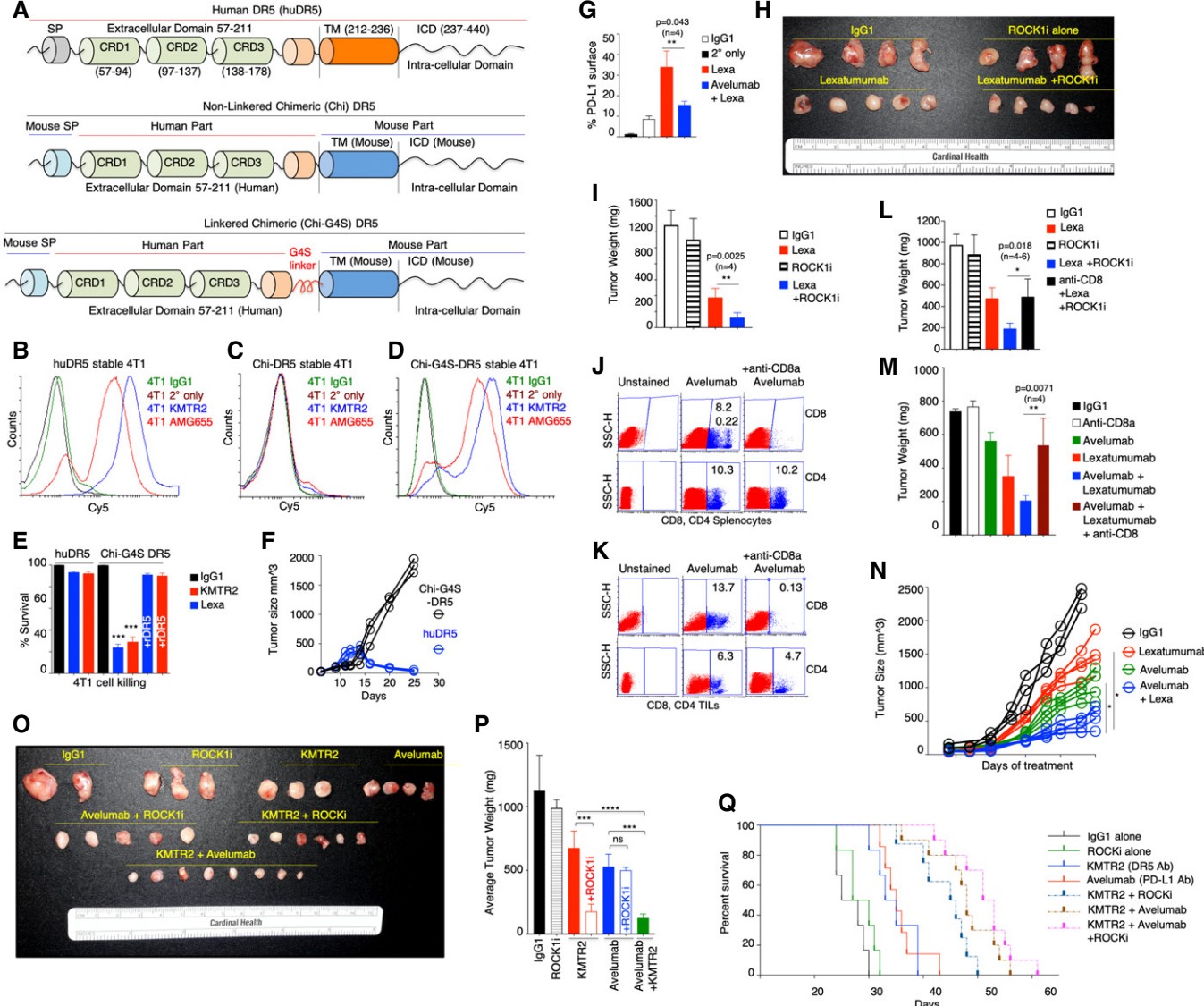

**Figure 5.**

**Figure 5. Generation and testing of chimeric DR5 for human DR5 agonist targeting in immunocompetent murine models.**

A Schematic and genetic construction of two chimeric human-mouse DR5 (Chi-DR5, Chi-G4S-DR5) with human extracellular domain and mouse transmembrane (TM) and intracellular domains (ICD).

B–D FACS plots confirming expression of human DR5 (huDR5) and Chi-G4S-DR5 in mouse 4T1 cells. Chi-DR5 was not expressed on cell surface (C).

E Cell viability analysis of huDR5 and Chi-G4S-DR5 stable 4T1 cells with indicated human DR5 agonists lexatumumab and KMTR2 ($n = 3$).

F Comparison of tumor growth of grafted huDR5 and Chi-G4S-DR5 stable 4T1 cells in BALB/c mice ($n = 3$).

G Chi-G4S-DR5 stable 4T1 tumors (after reaching ~150–200 mm$^3$) were treated with four doses (100 µg) of IgG1, lexatumumab, lexatumumab + avelumab followed by tumor recovery and surface PD-L1 analysis using flow cytometry similar to described in Fig 1I and J ($n = 3$–4). See also Appendix Fig S9.

H Chi-G4S-DR5 stable 4T1 tumors (after reaching ~100 mm$^3$) were treated with six doses of indicated treatment. Isolated tumors were imaged together ($n = 4$–5). ROCK1i indicates GSK269962A and was administered to animals via intra-tumor injections (2 mg/kg).

I Same as H, except average tumor weight is shown ($n = 4$).

J, K Confirmation of selective CD8$^+$ T cell population depletion in mouse spleen and grafted tumors after injecting animals with either anti-PD-L1 avelumab alone or avelumab + anti-CD8a antibody. CD4$^+$ T cells remained unchanged.

L Average of tumor weights (harvested at same time) from mice treated (i.p.) with IgG1, lexatumumab, ROCK1i, lexatumumab + ROCK1i, and anti-CD8$^+$ lexatumumab + ROCK1i ($n = 4$–6, 50 µg lexatumumab, 2 mg/kg ROCK1i, six doses). ROCK1i indicates GSK269962A and was administered to animals via intra-tumor injections. Various treatments were started when tumors were ~100 mm$^3$ size. Also see Fig EV4B.

M Average of tumor weights (harvested at same time) from mice treated (i.p.) with IgG1, lexatumumab, avelumab, lexatumumab + avelumab, and anti-CD8$^+$ lexatumumab + avelumab ($n = 4$–6).

N 6- to 8-week-old C57BL/6-bearing Chi-G4S-DR5-expressing MC38 tumors were intraperitoneally (i.p.) injected with 50 µg of indicated antibody every third day ($n = 4$–6). Various treatments were started when tumors were ~100 mm$^3$ size. Tumor volumes were quantified at indicated days by caliper measurements.

O Same as H, except KMTR2 (8 doses) antibody instead of lexatumumab was used as the DR5 agonist for the experiment.

P Average of tumor weights of data shown in (O).

Q Kaplan–Meier plot depicting the survival of syngeneic Chi-G4S-DR5 4T1 tumor-bearing animals injected i.p. with 100 µg of indicated antibodies such as IgG1, KMTR2, and avelumab. Animals were injected with GSK269962A 2 mg/kg (in PBS) directly into tumors wherever ROCK1i is indicated.

Data information: Mean ± SD. Statistical significance in (E), (G), (I), (L), (M), and (P) was determined by unpaired two-tailed *t*-test and in (N) using two-tailed paired Wilcoxon–Mann–Whitney test ($n = 4$–6; \*$P < 0.05$, \*\*$P < 0.005$, \*\*\*$P < 0.0001$, \*\*\*\*$P < 0.00005$).

Source data are available online for this figure.

been described the key prognostic predictor of effective anti-tumor response in TNBC and other tumors (Wang *et al*, 2017a). Strikingly, we observed a higher CD8$^+$/CD4$^+$ ratio in lexatumumab + ROCK1i and lexatumumab + avelumab-treated 4T1 tumors but not in DR5 agonist alone treated tumors (Fig 6C, Appendix S10, $n = 6$–20 tumor-bearing animals, three separate experiments). Similar results of significantly higher granzyme-b activity (a marker of cytotoxic T-cell function) were evident in avelumab, lexatumumab + ROCK1i, and lexatumumab + avelumab-treated size-matched tumors as compared to DR5 agonist alone treated chi-G4S-DR5 stable MC38 and 4T1 tumors (Figs 6D and EV4E). Interestingly, DR5 agonist co-treatments did not enhance activity of regulatory T cells (T-regs) in tumors as evident with Foxp3 Western blots and IHC data (Figs 6D, EV4E, and EV5). Collectively, these sets of comprehensive investigations collective support higher tumor breakdown and higher anti-tumor immune function of DR5 agonists when given in combination of ROCK1 inhibitors or PD-L1 function inhibiting antibodies.

### DR-5-PD-L1 co-targeting bispecific antibody for solid tumors

PD-L1 is a key regulator of immune suppression in TNBC and other solid cancers, and FDA has approved various PD-L1/PD-1 blocking antibodies (Aktas *et al*, 2019). As DR5 agonist stabilized PD-L1 on cell surface, we next hypothesize that elevated surface PD-L1 if used as an anchor will not only activate T cells but will also use PD-L1 as an anchor to enhanced DR5 signaling (Shivange *et al*, 2018). Thus, we next genetically engineered avelumab and murine DR5 agonist MD5-1 antibody into bispecific antibody called avelu-MD5-1 (Fig 6E and F), capable of not only blocking PD-L1 immunosuppressive function but also enhancing DR5-DISC clustering (Fig 6G). Both antibodies were confirmed to have binding against murine MC38 cells and 4T1 cells (Fig EV4F and G). MD5-1 antibody requires Fc crosslinking to activate cell death (Shivange *et al*, 2018) while

avelu-MD5-1 was effective in killing murine 4T1 and MC38 cells (Figs 6H and EV4H). When tested in 4T1 TNBC syngeneic tumor models, avelu-MD5-1 completely regressed tumors (Fig 6I and J). Both MD5-1 alone and avelumab alone only stabilized the tumors at similar dose (Fig 6I and J). Next, size-matched MD5-1, avelumab, and avelu-MD5-1 tumor lysates (after six doses) were analyzed for granzyme-b and caspase-3 activity (Fig 6K). Similar to previous results of lexatumumab (Fig 6D), MD5-1-treated tumor lysates had high CD4 and CD8 expression as compared to IgG1-treated tumors (Fig 6K). However, the granzyme expression in MD5-1 lysates was similar to IgG1-treated tumors, while both MD5-1 and avelu-MD5-1 lysates had elevated cleaved caspase-3 levels (Fig 6K). Next, we analyzed for intracellular IFN-γ expression (an indicator of cytotoxic T-cell activity) in enriched CD8 population using flow cytometry studies. We observed a significantly higher percentage (multiple fold) of IFN-γ-positive CD8 cells in MD5-1 + ROCK1i, MD5-1 + PD-1 inhibitor (BMS202) and avelu-MD5-1 bispecific treated tumors as compared to DR5 agonist (MD5-1) alone (Fig 6L and M, Appendix Fig S11A–C, $n = 7$–16 tumors, three independent experiments). Collectively, these results raise a high expectation of potentially significantly effective clinical response in TNBC patients by FDA approved anti-PD-L1 atezolizumab if given in combination of DR5 agonists (Aktas *et al*, 2019).

## Discussion

Over last few decades, the faithful translation of many preclinical studies into human clinical trials has largely relied on homogenous cellular and heterogeneous patient-derived tumor xenograft models. In current era of cancer immunotherapy, these tumor models present significant hurdles to fully understand the complex cancer immunotherapy strategies by antibodies that eliminate tumor cells

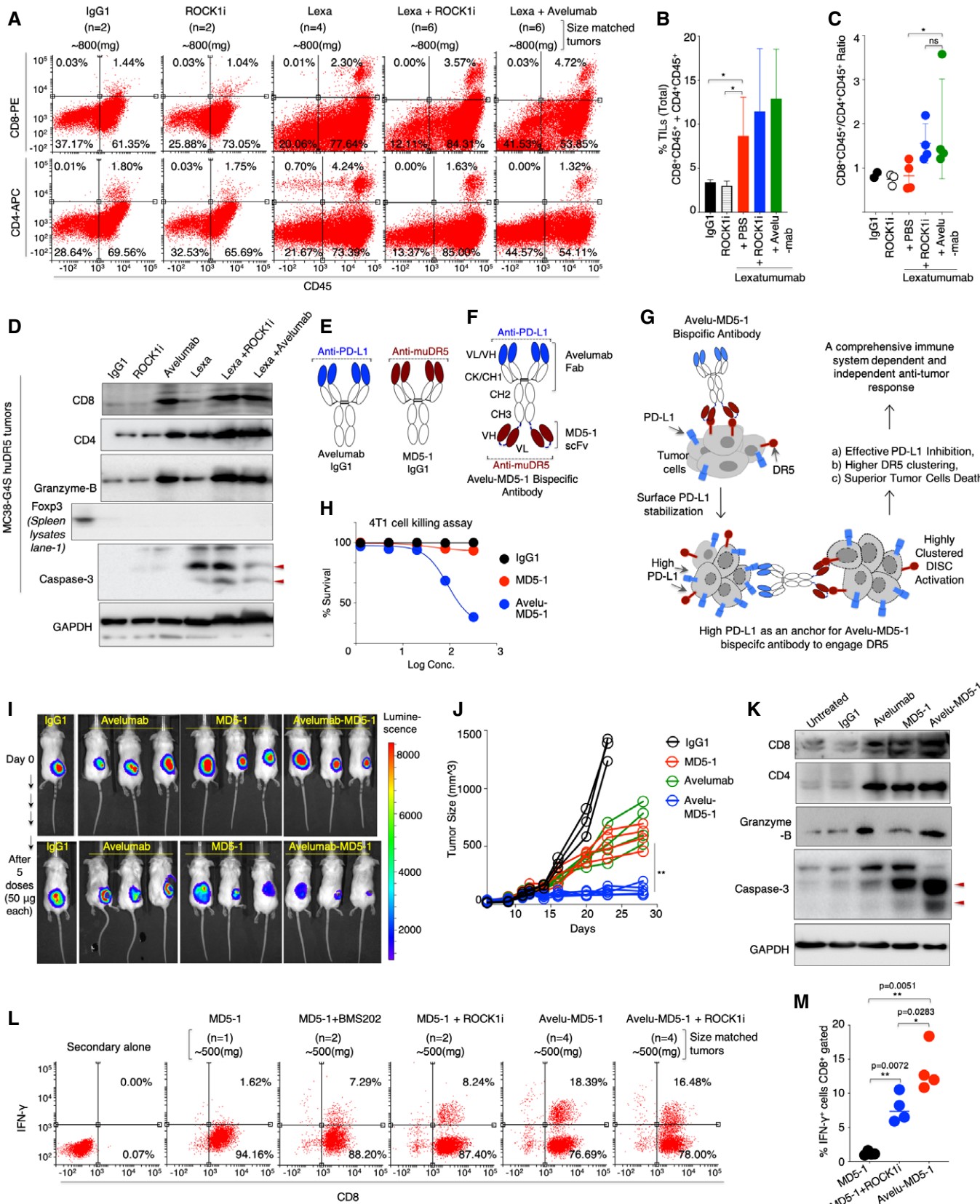

**Figure 6.**

**Figure 6.  Co-targeting of DR5 with ROCK1i or PD-L1 enhances immune infiltration, overpowers immune suppression, and improves anti-tumor activity.**

A  Chi-G4S-DR5 stable 4T1 tumors harboring mice were treated lexatumumab, lexatumumab + ROCK1i, and avelumab + lexatumumab and other controls as indicated. Antibodies were treated i.p at 100 µg dose (6 total), ROCK1i (in PBS) was injected directly into tumors at 2 mg/kg dose (6 total). Various treatments were started when tumors were ~400 mm³ size. Harvested tumors were grouped together and size-matched (3 independent sets: $n = 2$–6 tumors in each set) followed by TIL isolation (see methods). CD8/CD45 and CD4/CD45 expressing cells were measured by flow cytometry. The data shown are from a single set of experiment. For addition sets, see also Appendix Fig S10 ($n = 3$).

B  Plots showing % of total double positive TILs (CD8$^+$CD45$^+$ + CD4$^+$CD45$^+$) in right upper quadrant after combining three independent experiments. Indicated treatments are shown at the bottom of bars ($n = 3$).

C  Ratio of CD8$^+$CD45$^+$/CD4$^+$CD45$^+$ isolated TILs from tumors in each indicated treatment ($n = 3$).

D  Similar to (A) Chi-G4S-DR5 stable MC38 tumors harboring mice were treated (6 total doses) lexatumumab, avelumab, ROCK1i, lexatumumab + ROCKi, and avelumab + lexatumumab and IgG1 control as indicated. Harvested tumors homogenized followed by quantitation. Protein lysates were run on SDS–PAGE followed by immunoblotting using indicated CD8, CD4, Foxp3, caspase-3, and granzyme-b antibody. GAPDH is loading control. Red arrows indicated cleaved caspase-3 p19 and p17 fragments. For additional Chi-G4S-DR5 stable 4T1 tumors western data, see Fig EV4E.

E  Schematic of anti-PD-L1 (avelumab) and anti-muDR5 (MD5-1) IgG1 antibodies. Highlighted blue area depicts avelumab's antigen binding variable domain, and dark red area indicates MD5-1's antigen binding variable domain.

F  Schematic and genetic construction of avelu-MD5-1 bispecific antibody that contains anti-PD-L1 (avelumab) and anti-muDR5 (MD5-1) heavy/light (VH/VL) variable domains (blue and dark red), respectively. Both monospecific and bispecific antibodies contain LALA mutation to avoid interference with Fc-effector function. CK: C-kappa, CH1: Constant heavy chain 1, CH2: Constant heavy chain 2, CH3: Constant heavy chain 3. Dark blue dotted line under CH3 and between MD5-1 VH and VL depicts flexible linker.

G  Working mechanism of avelu-MD5-1 bispecific antibody where surface stabilized PD-L1 acts as an anchor to enhance avidity optimized binding and clustering of DR5 receptor-mediated apoptotic signaling.

H  Cell killing assay of 4T1 cells treated with murine DR5 agonist MD5-1 and bispecific avelu-MD5 antibody.

I  BALB/c mouse harboring luciferase stable 4T1 tumors were i.p. injected with indicated antibodies. 50 µg dose and mice were imaged after five doses.

J  6- to 8-week-old C57BL/6 mouse bearing MC38 tumors were intraperitoneally (i.p.) injected with 50 µg of indicated antibody every third day ($n = 4$–6). Indicated treatments were started when tumors were ~100 mm³ size. Tumor volumes were quantified at indicated days by caliper measurements.

K  ~150–200 mm³ size MC38 tumor-bearing C57BL/6 mice were treated with indicated MD5-1, avelumab and bispecific antibodies along with control IgG1, six total doses. Harvested tumors were homogenized followed by quantitation. Protein lysates were run on SDS–PAGE followed by immunoblotting using indicated CD8, CD4, caspase-3, and granzyme-b antibody. GAPDH is loading control. Red arrows indicated cleaved caspase-3 p19 and p17 fragments.

L  6- to 8-week-old C57BL/6 mice were injected with MC38 cells. When tumors reached ~150–200 mm³, animals were intraperitoneally (i.p.) injected with 50 µg of indicated antibody every third day. On day 18, tumors were harvested, size-matched and pooled by treatment group, and exposed to collagenase/DNase and were single-cell suspensions enriched for CD8$^+$ cells. Enriched CD8$^+$ T cells from various treatments were restimulated with anti-CD3 (OKT3) antibody for four additional hours. CD8-gated cells were next analyzed for IFN-γ intracellular expression using flow cytometry. The data shown are from a single set of experiment. See also Appendix Fig S11 ($n = 3$).

M  Percentage of IFN-γ$^+$CD8$^+$ double positive cells from three independent experiments. For supporting flow cytometry data, see also Appendix Fig S11 ($n = 4$).

Data information: Mean ± SD. Statistical significance in (B) and (C) was determined by Mann–Whitney two-tailed test and in (J) using two-tailed paired Wilcoxon–Mann–Whitney test. Statistical significance in (M) was determined by unpaired $t$-test (*$P < 0.05$, **$P < 0.005$).
Source data are available online for this figure.

by apoptotic cytotoxicity. As immune activating or repressive function of human DR5 agonist antibodies remains largely untested and since all DR5 agonist have failed to move beyond phase-II trials (Ashkenazi, 2015), we undertook these investigations to underpin the potential immune inhibitory function of DR5 agonists if any.

Our finding with various solid tumor cell lines, multiple solid tumor xenografts, and using multiple clinical DR5 agonist antibodies support PD-L1 stabilization and immunosuppressive role of DR5 agonists. PD-L1 surface mobilization was also maintained regardless of EMT acquisition in tumor cells. Using an array of experimental, preclinical, and clinical DR5 agonist antibodies targeting mouse, human, and chimeric DR5, our findings have discovered an unexpected PD-L1-based immune evasion mechanism that is not selective to a particular DR5 antibody, which binds to a particular DR5 epitope. Rather, the immune evasion is driven by DISC-caspase-8 signaling. Consistent with previous reports, our results support caspase-mediated cleavage of proteasome complex (Cohen, 2005). Moreover, in agreement with previous reports of PD-L1 regulation by deubiquitinase CSN5 (Lim et al, 2016), proteasome inhibition and deubiquitinase activity serve the same purpose via different upstream mechanisms.

As lexatumumab, apomab, AMG655, and tigatuzumab have been tested clinically in gastric cancers, TNBC and other solid cancers (Kalthoff & Trauzold, 2009), future IHC and ISH studies from treated patient samples will strengthen the results in human. Unfortunately, we could not get access to DR5 agonist-treated patient samples.

Regardless, in grafted TNBC PDX tumors, we observed high PD-L1 in IHC studies after DR5 agonist treatments and their co-targeting (PD-L1 + DR5) significantly improved immune effector function and anti-tumor efficacy. Continuing on the similar note, an appropriate immune competent tumor model remains a challenge in DR5 field. Indeed, when tested using MMTV-PyVT TNBC cellular GEM tumor grafts (Usary et al, 2016), <10% cells stained positive for DR5 as compared to surrogate 4T1, MC38 syngeneic models (Appendix Fig S11D). Furthermore, these breast TNBC GEM models (MMTV-PyVT, MMTV-neu) express significantly low PD-L1 on tumor cells (Nolan et al, 2017) and do not respond effectively to PD-1/PD-L1 and anti-CTLA4 immune checkpoint therapies unless given in combination of adjuvants such as cisplatin (Nolan et al, 2017). Thus, described transgenic syngeneic tumor studies in this paper are very first report in the field to test human clinical DR5 agonists in immune competent tumor microenvironment.

Our findings of S5a proteasome 26s subunit degradation and ROCK1 activation by activated caspase-8 without amplification of caspase-3 and downstream cell death are intriguing as the cellular resistance due to Bcl-2 (Bcl-x) and other pro-survival proteins remains a critical clinical challenge to success of DR5 therapy (LeBlanc et al, 2002). In addition, various reports have described non-apoptotic function of caspase-8 during development (Miura, 2012; Solier et al, 2017), particularly its key role in T-cell development and immune homeostasis by another TNF superfamily Fas-ligand signaling pathway (Chun

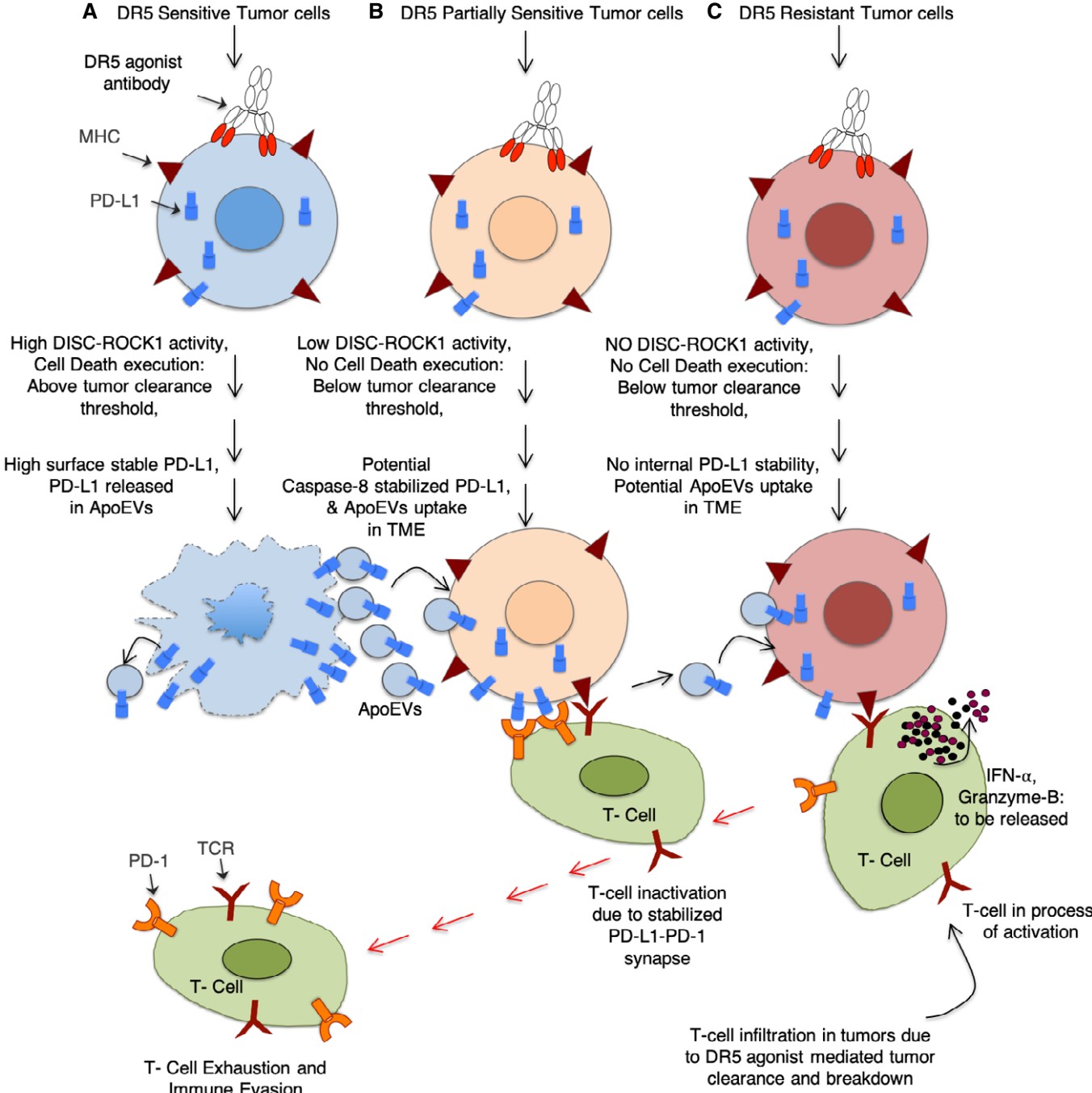

**Figure 7. Working model of PD-L1 immunosuppression by DR5 agonist antibodies.**

Heterogeneous tumors consisting of DR5-sensitive (A), partially sensitive (B) and potentially resistant (C) tumor cells. DR5 agonist activates cell death above tumor clearance threshold in sensitive cells. Activation of caspase-8 and caspase-3 inactivates proteasome function and stabilizes intracellular PD-L1. Activated ROCK1 potentially help mobilizes PD-L1 to membrane by some unknown mechanism, which is also released in ApoEVs from dying cells. These PD-L1-containing ApoEVs shuttle and transfer cargo PD-L1 to other heterogenous cell types in tumors (potentially DR5 resistant) to increase the overall basal pool of PD-L1 in tumors. At the same time, due to extrinsic DR5 agonist-mediated cytotoxicity, tumor cells are eliminated to generate partial tumor clearance and break down. However, incoming immune effector cells including T cells are exhausted in the tumors due to overactive PD-L1, thus, limiting their anti-tumor response. Co-targeting ROCK1-DR5 reduces ApoEVs stabilized PD-L1 pool in tumors, while anti-PD-L1-DR5 co-targeting reduces immunosuppressive function of both basal and ApoEVs stabilized PD-L1 in tumors.

*et al*, 2002; Salmena *et al*, 2003). Importantly, random caspase cascade activation by any other apoptotic agent that act through mitochondria are not involved in leukocyte development (Chun *et al*, 2002; Salmena *et al*, 2003). Furthermore, as caspase-8 is activated by both pro-survival and pro-death signals (Newton *et al*, 2019), the function of partially activated caspase-8 (by antibodies that activate apoptosis below tumor clearance threshold) in regulating proteasome activity and cytoskeleton without activating cell death indicates its differential regulation in cancer cells than immune cells (Ferrari *et al*, 1998). If clinical proteasome inhibitors induce immune suppression in patients is beyond the scope of described investigations.

Between rho binding domain (RBD) domain and pleckstrin homology (PH) domain, although ROCK1 contains two optimal caspase-8/caspase-3 cleavage XEXD sites, recent studies have established that either distal regions of substrate itself or other regulators in complex could optimally prefer caspase-8 over caspase-3 (Baker & Masters, 2018). Similarly, caspase-8 regulation by two different forms of cFLIP (large and small) has been described, which can result in entirely different outcome (Hughes *et al*, 2016). As Rho family of GTPases are key player in lymphocyte development and activation, and cancer cell exploiting one of their family member (ROCK1) to escape immune activity is similar to hijacking of macrophage-produced complement C1q to promote their own tumor growth (Roumenina *et al*, 2019).

Tumor cells having lower apoptotic threshold not only mobilized PD-L1 on cell surface (both *in vitro* and *in vivo*) but also shuttle it to neighboring tumor cells in a process that requires ROCK1 activation (Fig 7). ROCK-1 inhibitors that block metastasis and invasion are already in clinical trials (Chin *et al*, 2015). How ROCK1 help mobilize PD-L1 to cell surface demands further investigations. If activated ROCK1 makes use of same set of regulators that are important for its established membrane blebbing and cytoskeletal ruffling function (Coleman *et al*, 2001) or some other mechanisms that potentially also include other regulators such as CMTM6 (Fig 4) to help mobilize PD-L1 to tumor cell surface is beyond the scope of these investigations CMTM6 (Burr *et al*, 2017). Regardless, these findings of PD-L1 shuttling via ApoEVs are also is in agreement with

described studies of immunosuppression by exosomal PD-L1 (Chen *et al*, 2018). Importantly, besides avoiding immunosuppression, an effective DR5 agonist must also drive superior receptor clustering to eliminate both low and high DR5 expressing tumor cell in heterogeneous solid tumors (Ashkenazi, 2015). The observed dual high caspase-3 and IFN-γ activity (Fig 6) of a DR5 and PD-L1 co-engaging bispecific antibody (compare to MD5-1 or avelumab alone) is a key strategy and is in agreement with previous reports of higher order DR5 clustering by an anchored approach (Shivange *et al*, 2018). Furthermore, we consistently found that DR5 agonist co-targeting either with anti-PD-L1 or ROCK1i gave the higher immune infiltrations (T cells) in tumors; however, animals treated with DR5 + PD-L1 co-targeting had higher overall survival and significantly higher IFN-γ activity as compared to DR5 + ROCK1i. We believe the observed differences are due to distinct PD-L1 inhibition working mechanisms in the TME. The anti-PD-L1 antibody blocks both the basal and overall PD-L1 function in TME, while ROCK1 inhibition only regulates intra-cellularly stabilized PD-L1 shuttling to cell surface without changing basal surface PD-L1 levels in TME. Moreover, unlike syngeneic or transgenic surrogate tumors, differential contributing role of immune inhibitory cytokines from T-regs (Shevach, 2004) and MDSCs (Condamine *et al*, 2014) could account for additional discrepancies in the TME of different tumor types.

After a comprehensive trial in PD-L1⁺ patient population, atezolizumab (anti-PD-L1 therapy) was recently approved for metastatic TNBC patients expressing PD-L1 (Mavratzas *et al*, 2020), and since a large proportion of TNBC also express elevated DR5 levels (Forero-Torres *et al*, 2010), if a bispecific PD-L1-DR5 antibody will further improve survival in TNBC patients need to be seen in clinical trials. Given that key factor determining tumor progression within spatiotemporal dynamics, is the infiltration and activity of cytotoxic T cells in the TME (Binnewies *et al*, 2018) and considering most preclinical DR5 agonist studies have relied on xenograft models lacking T cells and TME induced immunological changes in tumors (Motoki *et al*, 2005; Zhang *et al*, 2007; Camidge, 2008; Kaplan-Lefko *et al*, 2010), our results support orchestration of an immune suppression by human and murine DR5 agonists.

# Materials and Methods

## Reagents and Tools table

| Reagent/Resource | Reference or Source | Identifier or Catalog Number |
|---|---|---|
| **Experimental Models (Cell lines)** | | |
| Human: OVCAR-3 | Ovarian Cancer | ATCC HTB-161 |
| Human: MDA-MB-436 | TNBC | ATCC HTB-130 |
| Human: MDA-MB-231 | TNBC | ATCC HTB-26 |
| Human: MDA-MB-231-2B | TNBC | ATCC HTB-26 |
| Human: PD-1 Effector Cells | Promega | J1151 |
| Human: PD-L1 aAPC/CHO-K1 Cells | Promega | J1091 |
| Human: A549 | Lung Cancer | ATCC CLL-185 |
| Human: Cavo-3 | Ovarian Cancer | ATCC HTB-75 |
| Human: HCC1806 | TNBC | ATCC CRL-2335 |
| Human: PANK1 | Pancreatic Cancer | ATCC CRL-1469 |

**Reagents and Tools table** (continued)

| Reagent/Resource | Reference or Source | Identifier or Catalog Number |
|---|---|---|
| Human: U87 | Brain Cancer | ATCC HTB-14 |
| Human: HCT116 | Colon Cancer | ATCC CCL247 |
| Human: Colo-205 | Colon Cancer | ATCC CCL-222 |
| Human: CHO-K cells | Stable transformed in our lab | ATCC CCL-61 |
| Mouse: 4T1 | Murine TNBC cell, Gift from Kevin Janes, UVA | ATCC CRL-2539 |
| Mouse: MC38 | Murine Colon Cancer, Kind gift from Dr. Suzanne Ostrand-Rosenberg, UMBC | CVCL_B288 |
| Mouse: ID8 | Murine Ovarian Cancer, Kind gift from Melanie Rutkowski | ABC-TC3940 |
| PDX cell line : UCD52 | Dr. Chuck Harrell | VCU Pathology core |
| Mouse: MC38 Chimeric human-mouse DR-5 G4S (Chi-G4S-DR5 cells) | Generated in our laboratory (This paper) | Human DR5 expressing murine colon cancer cells |
| Mouse: 4T1 Chimeric human-mouse DR5 G4S (Chi-G4S-DR5 cells) | Generated in our laboratory (This paper) | Human DR5 expressing murine TNBC cells |
| Mouse: 4T1 Chimeric human-mouse DR5 no G4S linker (Chi-DR5 cells) | Generated in our laboratory (This paper) | Human DR5 expressing murine TNBC cells |
| Mouse: MC38 Chimeric human-mouse DR5 no G4S linker (Chi-DR5 cells) | Generated in our laboratory (This paper) | Human DR5 expressing murine colon cancer cells |
| Mouse: 4T1 complete human DR5 no G4S linker (huDR5 cells) | Generated in our laboratory (This paper) | Human DR5 expressing murine TNBC cells |
| Mouse: ID8 Chimeric human-mouse DR5 G4S (Chi-G4S-DR5 cells) | Generated in our laboratory (This paper) | Human DR5 expressing murine ovarian Cancer cells |
| Human: MDA-MB-436 DR5-KO | Generated in our laboratory (This paper) | DR5 knockout cells |
| Human: MDA-MB-231 DR5-KO | Generated in our laboratory (This paper) | DR5 knockout cells |
| Human: MDA-MB-231-2B DR5-KO | Generated in our laboratory (This paper) | DR5 knockout cells |
| Human: U87-DR5-KO | Generated in our laboratory (This paper) | DR5 knockout cells |
| Human: MDA-MB-436 DR5 antibody Resistant | Generated in our laboratory (This paper) | DR5 resistant Cells |
| Human: MDA-MB-231 DR-5 antibody Resistant | Generated in our laboratory (This paper) | DR5 resistant Cells |
| Human: OVCAR3 DR-5 antibody Resistant | Generated in our laboratory (This paper) | DR5 resistant Cells |
| **Experimental Models (Mouse)** | | |
| Mouse: Athymic Nude Foxn1nu | Envigo | RRID:MGI:5652489 |
| Mouse: C57BL/6 | Jackson Lab | |
| Mouse: CD1(Crl:CD1(ICR) | Chalres River | |
| Mouse: SCID C.B-17/IcrHsd-Prkdc | Envigo, Dublin, VA | RRID:MGI:2160375 |
| Mouse: BALB/c | Jackson Lab | |
| Mouse: MMTV-PyMT | Jackson Lab | 2374 |
| **Antibodies** | | |
| Calreticulin | Cell Signaling Technology | Cat #12238 |
| Mouse Anti-Human CD47 | BD Biosciences | Cat #556044 |
| CSN5/COPS5 | Cell Signaling Technology | Cat #6895 |
| P-p65 | Cell Signaling Technology | Cat #3033 |
| p65 | Cell Signaling Technology | Cat #6956 |
| Caspase-8 | Cell Signaling Technology | Cat #9746 |
| P-MLC | Cell Signaling Technology | Cat # 3675 |
| E-Cadherin | Cell Signaling Technology | Cat #4065 |
| E-Cadherin | Cell Signaling Technology | Cat #14472 |
| N-Cadherin | Cell Signaling Technology | Cat #13116 |

**Reagents and Tools table**  (continued)

| Reagent/Resource | Reference or Source | Identifier or Catalog Number |
|---|---|---|
| Vimentin | Cell Signaling Technology | Cat #5741 |
| P-STAT3 | Cell Signaling Technology | Cat #9145 |
| STAT3 | Cell Signaling Technology | Cat # 12640 |
| ERK | Cell Signaling Technology | Cat # 9102 |
| Folate Receptor alpha Polyclonal Antibody (FOLR1) | Invitrogen | Cat # PA5-24186 |
| FOLR1 | R & D System | Cat# MAB5646 |
| Anti-Rabbit-HRP antibody | Cell Signaling Technology | Cat # 7074 |
| Anti-Mouse-HRP antibody | Cell Signaling Technology | Cat # 7076 |
| PD-L1 Antibody | Novus Biologicals | Cat # NBP1-76769 |
| Commercial MD5-1 (anti-Murine DR5) antibody | Abcam | Cat # ab171248 |
| CD8a Monoclonal Antibody (53-6.7), PE, eBioscience™ | Invitrogen | Cat # 12-0081-82 |
| CD4 Monoclonal Antibody (RM4-5), APC, eBioscience™ | Invitrogen | Cat # 17-0042-82 |
| Mouse CD25/IL-2R alpha Alexa Fluor® 488-conjugated Antibody | R & D Systems | Cat # FAB9164G |
| Cy5 conjugated Anti-Human IgG (H+L) | Jackson ImmunoResearch | Cat # 709-175-149 |
| Alexa Fluor® 488 AffiniPure Goat Anti-Rabbit IgG (H+L) | Jackson ImmunoResearch | Cat # 111-545-003 |
| Cy™5 AffiniPure Donkey Anti-Mouse IgG (H+L) | Jackson ImmunoResearch | Cat # 715-175-150 |
| Ubiquitin polyclonal antibody | Enzolifesciences | Cat # ADI-SPA-200 |
| K48-linkage Specific Polyubiquitin Antibody | Cell Signaling Technology | Cat # 4289 |
| FOX3P | Cell Signaling Technology | 4275 |
| CD8 | NOVUS Biologicals | NBP2-29475 |
| CD4 | NOVUS Biologicals | NBP1-19371 |
| CD8 | Invitrogen | 12-0081-82 |
| Granzyme B | Cell Signaling Technology | Cat # 4275 |
| BV421 Rat Anti-Mouse CD45 | BD Biosciences | 563890 |
| **Chemicals, enzymes and other reagents** | | |
| Z-VAD-FMK | Apex Bio | Cat # A1902 |
| Z-DEVD-FMK | Apex Bio | Cat # A1920 |
| GSK 269962A | TargetMol | Cat # T3518 |
| GSK 429286 | Tocris | Cat # 3726 |
| Y27632 | Apex Bio | Cat # A3008 |
| Y39983 | Sclleckchem.com | Cat # S7935 |
| Doxorubicin | Cayman Chemical Company | Cat # 15007 |
| Etoposide | Millipore | Cat # 341205 |
| MG-132 | Sigma | Cat # 474787 |
| Tunicamycin | MP Biomedicals | Cat # 150028 |
| Xenolight D- luciferin potassium salt | PerkinElmer | Cat # P/N 122799 |
| CHO free style Media | Thermo Fisher | Cat # 12651014 |
| HiTrap MabSelect Sure column | GE | Cat # 11003493 |
| Protein-A resin | Thermo Fisher | Cat # P153142 |
| HisPur Ni-NTA resin | Thermo Fisher | Cat # 88221 |
| HiPure Plasmid Maxiprep kit | Invitrogen | Cat # K21007 |

**Reagents and Tools table**  (continued)

| Reagent/Resource | Reference or Source | Identifier or Catalog Number |
|---|---|---|
| Endpoint Chromogenic LAL endotoxin assay kit | Lonza | Cat # 50-648U |
| AlamarBlue Cell viability reagent | Thermo Fisher | Cat # DAL1100 |
| MTT reagent | Thermo Fisher | Cat # V13154 |
| Infusion | Takara BioScience | STO344 |
| CHO CD efficient Feed B | Life Technologies | Cat # A1024001 |
| PEI transfection reagent | Thermo Fisher | Cat # BMS1003A |
| Matrigel | Corning | Cat # 354234 |
| Mouse anti-human IgG1 Fc | Thermo Fisher | Cat # A10648 |
| EZ-Link Sulfo-NHS-SS-Biotin | Thermo Fisher | Cat # 21331 |
| Ficoll®-Paque PREMIUM 1.084 | GE Healthcare | Cat # 17-5446-02 |
| Corning® 500 ml RPMI 1640 | Corning | Cat # 10-040-CV |
| Corning® 500 ml DMEM (Dulbecco's Modified Eagle's Medium) | Corning | Cat # 10-13-CV |
| VWR® Cell Strainers, DNase/RNase Free, Non-Pyrogenic, Sterile | VWR | Cat # 10199-656 |
| Halt protease inhibitor | Thermo Fisher | Cat # 78430 |
| AST reagent | Pointe Scientific | Cat # 23-666-1221 |
| EnzyChrom ALT Assay Kit | Bioassay Systems | Cat # EASTR-100 |
| Goat anti-Human IgG (H&L) Coated Magnetic Particles, Smooth Surface | Spherotech | Cat # HMS-30-10 |
| Superscript II | Invitrogen | Cat # 18064014 |
| BMS202 | Abcam | Cat# 231311 |
| **Software** | | |
| Vector NTI | Thermo scientific | N/A |
| GraphPad Prism | GraphPad Software | www.graphpad.com |
| FlowJo | FlowJo, LLC | www.flowjo.com |
| FCS Express | De Novo Software | www.denovosoftware.com |
| **Recombinant DNA** | | |
| Full Length Lexatumumab IgG and scFv | This Paper | GeneArt, Thermo Fisher |
| Full Length Farletuzumab IgG and scFv | This Paper | GeneArt, Thermo Fisher |
| Full Length AMG-655 IgG and scFv | This Paper | GeneArt, Thermo Fisher |
| Full Length KMTR2 IgG | This Paper | GeneArt, Thermo Fisher |
| Full Length MD5-1 IgG and scFv | This Paper | GeneArt, Thermo Fisher |
| Full Length AMG655 IgG | This Paper | GeneArt, Thermo Fisher |
| Full Length Tigatuzumab IgG | This Paper | GeneArt, Thermo Fisher |
| Full Length avelumab IgG and scFv | GeneArt, Thermo Fisher | GeneArt, Thermo Fisher |
| Full Length anti-CD3 IgG | This Paper | GeneArt, Thermo Fisher |
| Full Length Recombinant DNA, huFOLR1-IgG4-Fc | This Paper | GeneArt, Thermo Fisher |
| Full Length Recombinant DNA, huDR5-IgG4-Fc | This Paper | GeneArt, Thermo Fisher |
| Full Length Idarucizumab IgG and scFv | GeneArt, Thermo Fisher | GeneArt, Thermo Fisher |
| pcDNA3.1 | Thermo Fisher | V79020 |
| pET-28a | Addgene | 69864-3 |
| pTT5 | Durocher and Butler (2009) | Addgene (52326) |
| Lentiviral Vector chi-G4S-DR5 | System Biosciences | CD530-2 |
| Lentiviral Vector chi-G4S-DR5 | System Biosciences | CD550A-1 |

## Methods and Protocols

### Mouse strains

Six- to 8-week-old (age), 20–25-g (weight), both male and female (sex) mice were used for tumor xenografts generation, *in vivo* efficacy studies, imaging studies, TIL isolation studies as described in the text. Following mouse strains were used: C57BL/6 (Jackson laboratories), BALB/c (Jackson laboratories), immunodeficient BALB/c-derived athymic Nude Foxn1$^{nu}$/Foxn1$^{+}$ (envigo), and NOD.Cg-Prkdc$^{scid}$ Il2rg$^{tm1Wjl}$/SzJ also called as NSG mice. All animal procedures were conducted under the accordance of University of Virginia Institutional Animal Care and Use Committee (IACUC) approved protocol and conform to the relevant regulatory standards.

### Cell lines

The cell lines used in the study are provided in Table S2. All the cell lines were maintained in DMEM, MEM, RPMI-1640 or other required optimal medium supplemented with 10% heat-inactivated fetal bovine serum (FBS), 2 mM glutamine, 100 U/ml penicillin, and 100 μg/ml streptomycin (complete medium) unless otherwise specified as described (Shivange *et al*, 2018). MC38 cells (provided by S. Ostrand-Rosenberg, University of Maryland) were cultured in DMEM supplemented with 10% (vol/vol) FCS and 1 mM penicillin/streptomycin. Patient-derived cell lines were maintained in 20% FBS and 100 mM sodium pyruvate in RPMI 1640 media supplemented with glutamax (Gibco) and 1% penicillin/streptomycin (Gibco). Various cell lines were trypsinized and expanded as follows: After digestion, the cell suspension was neutralized with complete media and centrifuged 5 min at 1,500 rpm. The cell pellets were suspended in relevant DMEM/RPMI media and either expanded or seeded after counting using countess II (Life Technologies). Passaged cell lines were routinely tested for mycoplasma using MycoAlert Detection Kit (Lonza).

### Recombinant antibody cloning

The sequence of various clinical DR5 agonist antibodies is provided in Table S3. All clinical antibodies were clones and expressed as described earlier (Shivange *et al*, 2018). Antibodies were engineered by genetically linking variable regions of farletuzumab (anti-FOLR1 antibody) and lexatumumab (anti-TRAIL-R2/DR5 antibody) into human IgG1 framework. The DNA sequences were retrieved from the open sources (IMGT.ORG or publically available patents) and synthesized as gene string using Invitrogen GeneArt gene synthesis services. After PCR amplification using common gene string primers (forward primer: CTTTCTCTCCACAGGTGTCCACTC, reverse Primer: TTCCTTTATTAGCCAGAGGTCGAGGTC for all antibody cloning purposes), DNA was gel purified and inserted into pcDNA3.1 vectors (CMV promoter) by making use of In-Fusion HD Cloning Kits (Takara Bio). EcoR1 and HindIII digested vector was incubated with overlapping PCR fragments (of various different recombinant DNAs, see list of clones in Reagent Table) with infusion enzyme (1:2, vector: insert ratio) at 55°C for 30 min, followed by additional 30 min incubation on ice after adding *E. coli* Stellar™ cells (Clontech, see Reagent Table). Transformation and bacterial screening were carried out using standard cloning methods. Positive clones were sequenced confirmed in a 3-tier method. Confirmed bacterial colonies were Sanger sequencing upon PCR followed by re-sequencing of mini-prep DNA extracted from the positive colonies. Finally, maxiprep were re-sequenced prior to each transfection. Recombinant antibodies were also re-confirmed by ELISA and flow cytometry surface binding studies as described earlier (Shivange *et al*, 2018).

### Recombinant antibody expression

Free style CHO-S cells (Invitrogen, Reagent Table) were cultured and maintained according to supplier's recommendations (Life Technologies) biologics using free style CHO expression system (Life Technologies) and as previously described (Durocher & Butler, 2009; Shivange *et al*, 2018). A ratio of 1:2 (light chain, VL: heavy chain, VH) DNA was transfected using 1 μg/ml polyethylenimine (PEI, see Reagent Table). After transfection, cells were kept at 37°C for 24 h. After 24 h, transfected cells were shifted to 32°C to slow down the growth for nine additional days. Cells were routinely feed (every 2$^{nd}$ day) with 1:1 ratio of Tryptone feed and CHO Feed B (see Reagent Table). After 10 days, supernatant from cultures was harvested and antibodies were purified using protein-A affinity columns. Various recombinant antibodies used in this study and recombinant target antigens were engineered, expressed and purified in Singh Laboratory of Novel Biologics as described earlier (Shivange *et al*, 2018). Recombinant human Apo2L/TRAIL was obtained from R&D systems. His-tag Apo2L was also expressed and purified using nickel NTA columns (see Reagent Table) using standard BL21 bacterial expression system. His-Apo2L generated in our laboratory was confirmed (alongside of commercial Apo2L) using multiple cancer lines. Similarly, the activity of commercial MD5-1 antibody was compared next to recombinant MD5-1 generated in our laboratory using multiple cancer cell lines as described earlier (Shivange *et al*, 2018).

### Antibody purification

Various transfected monospecific and bispecific antibodies (as indicated in text and figure legends) were affinity purified using HiTrap MabSelect SuRe (GE, 11003493) protein-A columns. Transfected cultures were harvested after 10 days and filtered through 0.2-μm PES membrane filters (Millipore Express Plus). Cleaning-in-place (CIP) was performed for each column using 0.2 M NaOH wash (20 min). Following cleaning, columns were washed three times with binding buffer (20 mM sodium phosphate, 0.15 M NaCl, pH 7.2). Filtered supernatant containing recombinant antibodies or antigens was passed through the columns at 4°C. Prior to elution in 0.1 M sodium citrate, pH 3.0–3.6, the columns were washed three times with binding buffer (pH 7.0). The pH of eluted antibodies was immediately neutralized using sodium acetate (3 M, pH 9.0). After protein measurements at 280 nm, antibodies were dialyzed in PBS using Slide-A-Lyzer 3.5K (Thermo Scientific, 66330). For experimentation, antibodies were diluted in 1× PBS and indicated concentrations (see main text and figure legends) were used for indicated treatments. Antibodies were run on gel filtration columns (next section) to analyze the percent monomers. Whenever necessary a second step size exclusion chromatography (SEC) was performed, and antibodies were always diluted in 1× PBS. Recombinants IgG4-Fc tagged extracellular domain antigens such as rFOLR1, rDR5 etc. were also similarly harvested and purified using protein-A columns and diluted in 1× PBS prior to indicated treatments.

### Size exclusion chromatography

The percent monomer of purified antibodies was determined by size exclusion chromatography. 0.1 mg of purified antibody was injected into the AKTA protein purification system (GE Healthcare Life Sciences), and protein fractions were separated using a Superdex 200 10/300 column (GE Healthcare Life Sciences) with 50 mM Tris (pH 7.5) and 150 mM NaCl. The elution profile was exported as Excel file, and chromatogram was developed. The protein sizes were determined by comparing the elution profile with the gel filtration standard (Bio-Rad 151-1901; Hong et al, 2012). Any protein peak observed in void fraction was considered as antibody aggregate. The area under the curve was calculated for each peak, and a relative percent monomer fraction was determined as described earlier (Shivange et al, 2018).

### Binding studies by ELISA

Binding specificity and affinity of various described IgG1 subclasses were determined by ELISA using the recombinant extracellular domain of DR5/TRAIL-R2. For coating 96-well ELISA plates (Olympus), the protein solutions (2 μg/ml) were prepared in coating buffer (100 mM Sodium Bicarbonate pH 9.2) and 100 μl was distributed in each well. The plates were then incubated overnight at 4°C. Next day, the unbound areas were blocked by cell culture media containing 10% FBS, 1% BSA and 0.5% sodium azide for 2 h at room temperature. The serial dilutions of antibodies (2-fold dilution from 50 to 0.048 nM) were prepared in blocking solution and incubated in target protein-coated plates for 1 h at 37°C. After washing with PBS solution containing 0.1% Tween-20, the plates were incubated for 1 h with horseradish peroxidase (HRP)-conjugated anti-human IgG1 (Thermo Scientific, A10648). Detection was performed using a two-component peroxidase substrate kit (BD Biosciences), and the reaction was stopped with the addition of 2N sulfuric acid. Absorbance at 450 nm was immediately recorded using a Synergy Spectrophotometer (BioTech), and background absorbance from negative control samples was subtracted. The antibody affinities (Kd) were calculated by non-linear regression analysis using GraphPad Prism software.

### In vitro cell viability assays

Cell viability following Apo2L, lexatumumab, KMTR2, tigatuzumab, AMG-655, MD5-1, avelu-MD5, etc. treatments (as indicated in various figures) either alone in combination with inhibitors (ROCK1 or ERK or caspase inhibitors etc) or in combination with an anti-Fc reagent (for tigatuzumab, AMG-655, and MD5-1) was determined using the AlamarBlue cell viability assays and MTT cell proliferation assays as per manufactured protocols. Briefly, cells (indicated cells in main text or figure legends) were treated with increasing concentration of various antibodies (as indicated) along with relevant positive and negative control antibodies for 6, 24, or 48 h (as indicated according to the experiment). For each cell killing assay, the figures show the representative profiles from $n = 2$–4 with different cultured confluency. Whenever used for immunoblotting, following antibodies treatment caspase-3 processing in tumor cells was monitored using selective antibodies that recognize cleaved human caspase-3 or total caspase-3 (Cell signaling, 9661 and 9668). TRAIL-R2 receptor in oligomerization was determined using immunoblotting assays (cell signaling Rabbit mAb, 8074). Cell viability was additionally examined by flow cytometry based apoptotic detection methods using 7-aminoactinomycin D (7-ADD) exclusion from live cells.

### $IC_{50}$ determination

$IC_{50}$ values were calculated using MTT assays. Cells were seeded in 96-well plates. Next day, when cultures became adherent, cells were incubated for 48 h at 37°C (5% $CO_2$) with the increasing concentrations of the antibodies or drug (such as cisplatin) as indicated in experiments. Before treatments, various antibodies were dialyzed into PBS and typically had a pH around 7.5. Values obtained after reading the 96-well plates were normalized to IgG control antibody control, and $IC_{50}$ values were calculated using non-linear dose–response regression curve fits using GraphPad Prism software. The final results shown in the histograms were obtained from three independent experiments. Whenever provided in the curves, the error bars show ±SEM.

### Western blotting

Cells were cultured overnight in tissue culture-treated 6-well plates prior to treatment. After antibody treatment for 48 h (or indicated time), cells were rinsed with PBS and then lysed with RIPA buffer supplemented with protease inhibitor cocktail (Thermo Scientific). Spinning at 14,000 rpm for 30 min cleared lysates, and protein was quantified by Pierce BCA protein assay kit. Western blotting was performed using the Bio-Rad SDS–PAGE gel system. Briefly, 30 μg of protein was resolved on 10% Bis-Tris gels and then transferred onto PVDF membrane. Membranes were blocked for 1 h at room temperature in TBS + 0.1% Tween (TBST) with 5% non-fat dry milk. Membranes were probed overnight at 4°C with primary antibodies. All primary antibodies were used as 1:1,000 dilutions except GAPDH and tubulin which were used at 1:5,000 dilutions. Membranes were washed three times in TBST and then incubated with anti-rabbit or anti-mouse secondary antibodies (1/10,000 dilution, coupled to peroxidase) for 1 h at room temperature. Membranes were then washed three times with TBST, and immunocomplexes were detected with SuperSignal West Pico Chemiluminescent Substrate (Thermo Fisher Scientific). Images were taken using a Bio-Rad Gel Doc Imager system. Primary antibodies are listed in the Reagent Table.

### Biotinylation assay for PD-L1 surface expression

Cells were cultured as described earlier of this section. For this study, cells were grown in 10 cm culture dishes as monolayers and treated as mentioned in the text & figure legends of this manuscript. Treatment media was removed, and cells were washed twice with 12 ml of ice-cold PBS. After that, 15 ml of an ice-cold reaction solution of Sulfo-NHS-SS-biotin in PBS (250 μg/ml; $4.1 \times 10^{-4}$ M) was applied to each reaction and placed them on the ice-cold stainless-steel tray. Reactions were incubated for 10 min with gently rocking platform. Reactions were quenched by adding 10 ml of Tris–HCl solution per reaction. After removing cells from culture dish, surface was washed once with additional 5 ml of oxidized glutathione solution and this washing was collected with cells. Cells were collected by centrifugation in a refrigerated benchtop centrifuge and washed two times with 10 ml of glutathione containing PBS. Finally, lysis buffer was added into the cell pellet to make protein lysate. Protein concentration was measured. A 500 μl protein aliquot was taken in a fresh tube and 72 μl prewashed streptavidin-sepharose resin was

added. After 2 h of incubation at room temperature with occasional mixing by manual inversion of the tubes, the slurry was transferred into an Ultrafree centrifugal filter and centrifuged for 30 s at 16,100 $g$ in a centrifuge. The resin was washed three times with 400 μl of buffer A (Buffer A: 1% NP-40, 0.1% SDS in PBS), followed by two additional washings with 400 μl of buffer B (Buffer B: 0.1% NP-40, 0.5 M NaCl). The resin was quantitatively recovered from the filtering unit by means of two washing steps (each 50 μl) with PBS and transfer to a fresh tube. Centrifugation of the resin was carried out for 30 s at 16,100 $g$ in the centrifuge. Elution of protein was done by adding 500 μl of the elution solution (Elution solution: 6 M urea, 2 M thiourea, 30 mM D -biotin (Sigma), 2% SDS in PBS, pH 12.0) to the pelleted resin, resuspended by flicking the tube gently with fingertips, and incubated for 15 min at room temperature. Tubes were transferred to a 96°C heating block for 15 more minutes with occasional agitation. Finally, quantitatively the tube content was transferred into a new Ultrafree centrifugal filter and centrifuged for 30 s at 16,100 $g$ in a centrifuge, and the filtrate was collected and saved. SDS–PAGE was performed as described earlier.

### Pre-neutralization assays

Whenever indicated throughout the manuscript text or in figure legends, variable domain pre-neutralization of DR5 agonist antibodies or TNFα antibody was carried out. For *in vitro* and *in vivo* studies, indicated antibodies and indicated recombinant antigens (rDR5 etc.) were incubated together (either 1:1 or 1:5 ratio, as indicated) at 37°C for 1 h shaking on a platform. As a control, indicated non-pre-neutralized antibodies were also incubated at 37°C for 1 h shaking on a platform either with PBS alone or with recombinant non-specific proteins such as rHER2 or rGFP. Following pre-neutralization antibodies were either used *in vitro* for cell killing assays, or for cellular/tumor lysates generation (immunoblotting), or for live *in vivo* live imaging etc. as indicated.

### Flow cytometry

The cell surface expression of PD-L1 (1:500), CD47 (1:300), huDR5 (1:400), muDR5 (1:300), CD8 (1:300), CD4 (1:300), CD45 (1:500), CD25 (1:300) etc. was analyzed by flow cytometry. Overnight grown tumors were trypsinized and suspended in FACS buffer (PBS containing 2% FBS). The single-cell suspension was then incubated with primary DR4/DR5 antibodies for 1 h at 4°C with gentle mixing. Following wash with FACS buffer, the cells were then incubated with fluorescently labeled anti-Rabbit antibody for 1 h. Cells were washed and flow cytometry was performed using FACSCalibur. The data was analyzed by FCS Express (De Novo Software) and FlowJo.

### Generation of DR5-resistant cell lines

DR5-resistant TNBC and ovarian cell lines were generated in manner similar to described here (Wu *et al*, 2005) Anti-DR5 antibody resistance variants MDA-MB-436 and OVCAR3 of each cell line were derived from each original cell line by continuous exposure to antibodies following initial dose–response studies of KMTR2 and lexatumumab (0.1–100 nM) over 90 days. Cell viability assays were carried every week to test the resistance. Initially, each cell line was treated with 1 nM of lexatumumab for 72 h. The media was removed, and cells were allowed to recover for a further 48 h. Then, next round of treatment was carried out after doubling the previous treatment dose. The dose incremental analysis was carried out for approximately 3 months

for each cell line, during which IC50 concentrations were re-assessed in each resistant line. Cells were then maintained continuously in the presence of lexatumumab at these new IC50 concentrations for a further 2 months. After 3 months, resistance was confirmed using multiple DR5a agonists. Using this procedure, stable anti-DR5-resistant lines were generated for OVCAR3, MDA-MB-436, MDA-MB-231, etc.

### Epithelial-to-mesenchymal transition (EMT) of tumor cells

Epithelial-to-mesenchymal transition in indicated tumor cell lines (A549, OVCAR-3 etc) was induced as described earlier (Asiedu *et al*, 2011). Briefly, cells were treated with recombinant growth factors such HGF (c-Met), TNFα and TGFβ. Either Fc-conjugated or commercial c-Met, TNFα, and TGFβ were added to epithelial cell cultures A549 and OVCAR-3 with concentration of 50 ng/ml c-Met, 200 ng/ml of TNFα along with 60 ng/ml of TGFβ. These concentrations were standardized in laboratory after testing 20–200 ng/ml of TNFα and 10–10 ng/ml of TGFβ by keeping c-Met concentration constant. Treatment was carried out for a period of 4–6 days even after splitting the cells. Expression of E-cadherin, N-cadherin, and vimentin was analyzed after 24, 48, 72, and 96 h of treatment.

### Mechanical dissociation of tumors to obtain single-cell suspensions

Viable single cells from tumor tissues were isolated as described here (Leelatian *et al*, 2017). Briefly, after indicated antibody treatments (4–6 doses) mice were euthanized and tumors were harvested using sterile scissors and forceps. After excision of tumor, they were minced into small pieces in sterile RPMI-1640 media using two single-edged razor blades. Small tumor pieces were passed through a 70 μm cell strainer in sterile RPMI-1640 media. A rubber plunger and syringe were used to mesh the dissociated cells through the cell strainer and media containing dissociated cells was collected onto a sterile labeled conical tube. Dissociated tumor cells were subjected to flow cytometry (FACS) analysis for PD-L1, DR5, N-cadherin, FOLR1, CD47, etc. surface expression as described under flow cytometry protocol.

### PD-L1 surface expression analysis of tumor-derived cells

The cell surface expression of PD-L1 from tumor-derived cell was analyzed by flow cytometry. Isolated cells were suspended in FACS buffer (PBS containing 1% FBS). The single-cell suspension was then incubated with primary PD-L1 antibodies (1:400 dilution) for 1 h at 4°C with gentle mixing. We confirmed surface PD-L1 with commercially available antibodies as well as with clinical antibodies avelumab and atezolizumab (Zhang *et al*, 2017). Following three times wash with FACS buffer, the cells were then incubated with fluorescently labeled (Alexa 488) anti-Rabbit secondary antibody for 1 h. Cells were then washed 3–4 times with FACS buffer, and flow cytometry was performed using BD FACSCalibur. The data were analyzed by FCS Express (De Novo Software) and FlowJo software.

### HPG incorporation and analysis

Homopropargylglycine incorporation was essentially carried out as described earlier (Calve *et al*, 2016). HPG (Click Chemistry Tools) were diluted in PBS, raised to pH 7.4 with NaOH, sterilized with a 0.22 μm filter and stored at −20°C. Methionine-free RPMI media was purchased and was added to 100 nM final concentration of HPG. Cells were grown either in regular RPMI (called Met$^+$ media)

or in methionine-free RPMI with HPG (called Met⁻ HPG⁺ media). HPG incorporated proteins within the lysates were labeled selectively with Copper-catalyzed AF555-conjugated alkyne or azide using CuAAC, which results in a stable triazole adduct. The Click-iT® HPG Alexa Fluor 488 Protein Synthesis Assay Kit was used for flow cytometry as per manufacture protocol. Similar to $^{35}$S-methionine, Click-iT HPG is added to cultured cells and the amino acid is incorporated into proteins during active protein synthesis. Detection of the incorporated amino acid in cells for flow cytometry studies (following addition of exosomes or EVs) utilizes a chemoselective ligation or click reaction between an azide and alkyne, where the alkyne-modified protein is detected with either Alexa Fluor 488 or Alexa Fluor 594 azide. Labeled cells were analyzed for flow cytometry as described in main text.

### Exosomes and apoptotic cell-derived EV isolation

Exosomes were isolated as described here (Shen *et al*, 2011). For apoptotic cell-derived extracellular microvesicle (EV) isolation, clarified tissue culture supernatant was spun three times at 20,000 *g* for 30 min and was used. Although we only showed the data with apoptotic cell-derived EVs, similar trend in surface presence of PD-L1 was observed when purified exosome was added to DR5-KO cells.

For each trial, $6 \times 10^6$ cells were seeded onto 2 × 150 mm dishes in a total volume of 60 ml of DMEM supplemented with 10% exosome-free FCS and grown for 72 h. For all exosome studies, the tissue culture media was spun at 5,000 *g* for 15 min. The pellet was discarded, and the supernatants (SN) were passed through 0.22-μm filter. For exosome analysis by SPIRI and IFM, the filtrate was concentrated by angular flow filtration (Centricon Plus-70; EMD Millipore) to a final volume of to 500 μl. Exosomes were purified by size exclusion chromatography (Izon qEV column), 500 μl fraction samples were collected, and fractions 4, 5, and 6 were assayed by immunoblot (IB) to confirm the presence of exosome markers, pooled, and interrogated by SPIRI and IFM. For exosome analysis by IB, the clarified tissue culture supernatant was spun twice at 10,000 *g* for 30 min to remove contaminating microvesicles, and the resulting supernatant was spun at 70,000 *g* for 2 h at 4°C to pellet exosomes. Cell lysates were generated by addition of 2 ml of 2× SDS–PAGE sample buffer. Exosome pellets were resuspended in 600 μl of 2× SDS–PAGE sample buffer. Immunoblot at a constant ratio of exosome: cell lysates. IB analysis was performed by separating proteins by SDS–PAGE at a constant ratio of cell and exosome lysates. Proteins were then transferred to Immobilon membranes (EMD Millipore), followed by incubation with block solution (0.2% non-fat dry milk in TBST), primary antibody solution, and secondary antibody solution, with multiple washes in TBST between each step. Antigens were visualized by chemiluminescence and detected using an Amersham Imager 600 (GE Healthcare Life Sciences) gel imaging system. The resulting digitized IB images were then examined in ImageJ. Each was converted to 8-bit grayscale followed by background subtraction. Measurement parameter and scale were set to integrated density and pixel, respectively. Images were then inverted, bands were delineated using the freehand selection tool, and signal densities were converted to relative protein abundance by multiplying by the dilution factor for each sample. Relative budding was calculated by dividing the relative protein abundance in exosome lysate by the sum of the relative protein abundance in the cell lysate and the relative protein abundance in exosome lysate.

### Chimeric DR5 cloning and human DR5 agonist testing studies in BALB/c animals

For these investigations, we engineered chimeric DR5 that contained extra cellular domain (ECD) of human DR5 and transmembrane (TM) and intracellular domain (ICD) of mouse DR5. Two chimeric DR5 constructs were used: (i) without any linker between human ECD and mouse TM called Chi-DR5, and (ii) with G4S linker between human ECD and mouse TM called Chi-G4S-DR5. These constructs were synthesized using Invitrogen gene synthesis service and cloned in to either EF1α-driven viral expression vectors CD550A (pCDH-EF1α-MCS- BGH-PGK-GFP-T2A-Puro, system biosciences) or in pCDN3.1 vector with G418 selection and direct transfection using restriction-cloning sites (EcoR1- Not1). Viral infection had higher stable generation efficiency or direct lipid-based transfections. Only Chi-G4S-DR5 was expressed on 4T1 and MC38 cell surface as confirmed by FACs (see test in main manuscript). MC38 wild type cells and MC38 Chi-G4S-DR5 expressing lines were tested using C57BL/6J mice while 4T1 wild type 4T1 Chi-G4S-DR5 expressing lines were tested using BALB/c mice for syngeneic immune investigations. Since 4T1 is breast tumor lines, 6- to 8-week-old female BALB/c mice were injected with 4T1 Chi-G4S-DR5 cells subcutaneously (SC) in their right flank with $0.5 \times 10^6$ cells in Matrigel. These cells consistently formed tumors within ~2 weeks. Tumors were monitored as described previous section. Next, mouse bearing $\sim 100$ mm³ tumors were weight-matched and randomly assigned into groups and injected with antibodies as described in the text (100 μg of KMTR2 or lexatumumab alone or with 100 μg of avelumab). Similar tumor experiments were carried out using 4T1 WT cells and were treated with MD5-1 alone or MD5 plus avelumab. Antibodies were injected intraperitoneally. After 4–6 doses, tumors were harvested and subjected to tumor-infiltrating lymphocyte (TIL) isolation as described here (Tan & Lei, 2019) and in next section.

### TIL isolation by Ficoll-Paque density gradient centrifugation

TILs from 4T1 and MC38 tumors were isolated as described here (Tan & Lei, 2019). Briefly, after indicated antibody treatments mice were euthanized. Next, tumors were harvested using sterile scissors and forceps. After excision, tumors were minced into small pieces in RPMI-1640 media using two single-edged razor blades. Small tumor pieces were transferred to a 70 μm cell strainer in RPMI-1640 media. A rubber plunger of a syringe was used to mesh the dissociated cells through the cell strainer and cloudy media (that contained dissociated cells) was collected onto a sterile labeled 50-ml conical tube. Tubes were filled with 30 ml of RPMI-1640 media at room temperature (18–20°C). Immediately before the addition of Ficoll-Paque media, single-cell suspension was well mixed with 25-ml pipette. Thoroughly mixed 10 ml of Ficoll-Paque media was carefully poured in the bottom of the tube to form a layer of Ficoll-Paque below the cell suspensions without mixing the cell suspension. Tubes were centrifuged at 1,025 *g* for 20 min at 20°C with slow acceleration and without applying any brake. Twenty millilitre of the upper layer of media was discarded to a waste bottle from the tube. Layer of mononuclear cells that contained TIL was transferred to a sterile labeled 50-ml conical tube using a sterile pipette, along with the remainder of the media above the Ficoll-Paque. TILs were washed three times using 40 ml of complete RPMI media each time. After final wash with RPMI media, isolated tumor infiltrated

leukocytes (TILs) were subjected for flow cytometry (FACS) analysis directly or with anti-CD3 antibody (OKT3) incubation with fluorescently labeled CD4, CD8 and CD45, IFN-γ etc antibodies. antibodies as described in the text and here (Whitford *et al*, 1990).

### Immunohistochemistry (IHC) studies

All animal studies performed in here were in accordance with institutional guidelines at the Animal Care and Use Committee (ACUC) of the University of Virginia. To establish chimeric G4S DR-5 stable MC38 tumor (Chi-G4S-DR5), first $5 \times 10^5$ Chi-G4S-DR5 MC38 cells subcutaneously injected along with Matrigel in the right flank of 7- to 8-week-old female C57Bl/6 mice and allowed for tumor growth. Tumor growth was evaluated by measurement with calipers every 2 days. Tumors were measured in two perpendicular dimensions and the volume estimated using the formula [volume = 0.52 × (width)$^2$ × (length)] for approximating the volume (mm$^3$) of an ellipsoid. Treatment started when tumor volume reached at around 20 mm$^3$ volume. Mice were treated every alternate day with 100 μg of IgG1, avelumab, lexatumumab separately, and with combination of lexatumumab + avelumab and lexatumumab + ROCK inhibitor (GSK269962A, 2 mg/kg). Antibodies were injected intra peritoneally, while the ROCK1 inhibitor was injected near the tumors (intra-tumor) for six doses. Mouse tumors were collected at around 100–150 mm$^3$ volume & embedded in O.C.T. to make blocks and mouse spleen also collected & embedded similarly to use as a positive control. Samples were processed and sectioned into 4 μm tissue sections. Tumor sections were stained with CD4, CD8, and FOXP3 antibodies (1:150, 1:150, and 1:100 dilution was used, listed in the table) and counter-stained with hematoxylin. Peroxidase-conjugated anti-rabbit / anti-rat IgG reagents were used as secondary antibody. Reactions were developed using 3,3'-diaminobenzidine (DAB) as chromogenic substrate. Then, slides were dehydrated and mounted. Finally, brightfield images were taken using 50× objective lens of OLYMPUS BX41 microscope with OLYMPUS DP27 Camera. The scale bar in images Figs 1L and EV5 represents 50 and 20 μm, respectively. Only one representative tumor from each treatment was processed, and one representee image is shown.

Similarly, UCD52 TNBC PDX tumors treated either with six doses of KMTR2 or IgG1 (50 μg) were processed. In this case, tumor sections were stained with PD-L1 antibody (listed in the table) and counter-stained with hematoxylin. In this case, two different treated tumors were processed. Both showed higher PD-L1 stain. One representative image is shown in Fig 1L, while additional three images are shown in Appendix Fig S2C.

### ApoEV isolation

$4 \times 10^6$ cells were seeded onto 3 × 150 mm dishes in a total volume of 45 ml of DMEM supplemented with 10% exosome-free FBS and grown for 12 h. Media of each plate was changed with serum-free medium and incubated the cells at 37°C for 6–12 h, with or without various antibody treatments. For all ApoEV studies, the tissue culture media was spun at 8,000 *g* for 15 min. The pellet was discarded, and the supernatants (SN) were passed through 0.22-μm filter. ApoEVs were isolated by the ultracentrifugation method. Collected tissue culture media was spun at 5,000 *g* for 10 min to remove contaminating cellular particles, and the resulting supernatant was spun at 80,000 *g* for 4 h at 4°C to pellet ApoEVs. ApoEV pellets were resuspended in 500 μl of serum-free medium and were used for either cellular treatments, dot blot assays, and immunoblotting assays.

### Treatment of cells with ApoEV

$0.3 \times 10^6$ cells were seeded in 6-well plates day before analysis. Next day, media was changed with serum-free, ApoEV-free medium. After 12 h, again media was changed and 200 μl ApoEV suspension was added in each well of the plate and incubated for 24–96 h at standard culture condition.

### Dot blot assay

For dot blot analysis of ApoEVs, 30 μl of ApoEV suspension was heated at 95°C for 5 min. Then, protein dots were created on the Zeta-probe membranes using Bio-Dot® and Bio-Dot SF Microfiltration Apparatus. Blotting was carried out as per Bio-Rad immunoassay protocol.

### PD-1/PD-L1 reporter assay

PD1/PD-L1 reporter assay was carried with some modifications from manufacturer's original assay protocol (J1250 and J1255, Promega, (https://www.promega.com/products/reporter-bioassays/immune-checkpoint-bioassays/pd1_pdl1-blockade-bioassays/?catNum=J1250), similar to described here (Wang *et al*, 2017b). The original kit contains PD-1 effector Jurkat T cells stably express human PD-1 and NFAT-induced luciferase, while PD-L1 aAPC/CHO-K1 cells stably express human PD-L1 and a cell surface protein designed to activate cognate TCRs in an antigen-independent manner. Upon PD-1-PD-L1 interaction, luciferase signal is downregulated. Antibodies blocking PD-1-PD-L1 interaction removes inhibitory signals, resulting in luciferase activation. Modified assay for tumor-jurkat co-culture studies: MDA-MB-436, MDA-MB-231, A549, Colo-205, HCT-116 and OVCAR3 etc. cells were plated in 96-well cell culture plate at 40,000 cells/well in 100 μl of medium (RPMI/DMEM, 10% FBS, 0.2 mg/ml hygromycin-B) and incubated overnight at 37°C at 5% CO$_2$. Next day, medium was removed from the culture plate and cells were treated with diluted DR5 agonist antibodies. Antibodies were diluted in assay buffer containing RPMI-1640 + 1%FBS in 40 μl dilution per well, with or without inhibitors and incubated for few hours, as described in the text and figures. Glo response NFAT-luc2/PD1 Jurkat cells (Promega) were resuspended in assay buffer and incubated with 1 μg/ml anti-CD3 for 2 h to activate PD1 Jurkat cells. Activated PD1 cells were resuspended, and 40 μl of activated cells were added to each test well of 96-well culture plate (that contained DR5-treated tumor cells) at a concentration of $1.25 \times 10^6$ /ml. After 6 h of co-culture, assay plates were removed from the incubator; content of each well was mixed thoroughly with the help of multi-channel pipette and then transferred to a white opaque assay plate. Ten mg/ml luciferase reagent was prepared and loaded into the plate reader, which injects 50 μl reagent in each well and measures luminescence immediately. Data were analyzed using Microsoft excel software. Wherever indicated, inhibitors (ERK, STAT3, Caspase, ROCK1, etc.) were added 2–3 h prior to DR5 agonist antibodies in 96-well culture of tumors as described in the text and figures.

### Caspase-3 and Caspase-8 activity assay

Caspase-3 and caspase-8 activity was measured using Promega Caspase-Glo® 3/7 Assay kit (Cat No-G8091) and Caspase-Glo® 8 Assay kit (Cat No-G8201) following manufacturer's instruction. Around $1 \times 10^4$ cells were seeded in a 96-well plate and treated with various antibodies as described earlier of this manuscript. Before starting the assay, Caspase-Glo® 3/7 Buffer and lyophilized

Caspase-Glo® 3/7 Substrate were equilibrated to room temperature. The contents of the Caspase-Glo® 3/7 buffer bottle were transferred into the amber bottle containing Caspase-Glo® 3/7 substrate and mixed thoroughly to form the Caspase-Glo® 3/7 Reagent. 96-well plate containing treated cells also removed from the incubator and equilibrated to room temperature. Hundred microlitre of Caspase-Glo® 3/7 reagent was added to each well of a white-walled 96-well plate containing 100 µl of blank, negative control cells, untreated cells and treated cells along with culture medium. Plate was covered with a plate sealer. Contents of wells were gently mixed using a plate shaker at 300–500 rpm for 30 s and incubated at room temperature for 1 h. Luminescence of each sample was measured in a plate-reading luminometer. Similarly, caspase-8 activity was also monitored.

### Native PD-L1 immunoprecipitation studies with avelumab

Cells were cultured in 10 cm tissue culture dishes for 24 h prior to treatment. Before treatment, culture medium was replaced with serum-free medium. Cells were treated with indicated DR5 agonist antibodies (20–50 nM) as described in the text and figures. After 6 h of antibody treatment, 400 nM of avelumab (human anti-PD-L1 antibody with IgG4-Fc) was added to the culture media of each dish and incubated at 4°C for 1 h. Then, cells were harvested and lysed with IP lysis buffer (20 mM Tris pH 7.5, 150 mM NaCl, 1 mM EDTA, 10% Glycerol, 1% Triton-X, 0.5 mM PMSF) supplemented with protease inhibitor cocktail (Thermo Scientific). Spinning at 14,000 rpm for 30 min collected clear protein lysates, and protein was quantified by Pierce BCA protein assay kit. 1–1.5 mg protein (~400–500 µl) was taken into Eppendorf microcentrifuge tube. Protein lysates were incubated with anti-human IgG4-specific beads for 2 h at 4°C placing into a rotating wheel. Protein-conjugated beads were washed thrice with phosphate-buffered saline (PBS). Finally, beads were boiled at 100°C for 5 min with 30 µl of SDS sample buffer. 15–20 µl sample was loaded into the SDS gel, and Western blotting was performed using the Bio-Rad SDS–PAGE gel system followed by immunoblotting using indicated PD-L1-, CMTM6-, and ROCK1-specific antibodies.

### Generation of CRISPR DR5 Knockout tumor cells

Using Synthego's Gene Knockout Kit v2, we performed genomic deletion of DR5 in various tumor cell lines. Briefly, per manufacturer's instructions ribonucleoprotein (RNP) complexes that consist of purified Cas9 nuclease duplexed with chemically modified synthetic guide RNA (sgRNA) targeting DR5 were delivered to the cell lines by using Lipofectamine™ CRISPRMAX™ Transfection Reagent. DR5 knockout lines were further selected by treatment with 200 nM DR5 agonist antibody, which initiates apoptosis in left over wild-type cells to get a complete DR5 knockout population. In details, we first prepared plates by pre-warming 2 × 24-well cell culture plates with 500 µl of normal growth medium in each well. Next, RNP complexes were assembled in 1.3:1 sgRNA to Cas9 ratio and working concentrations (3 pmol/µl) of RNPs were prepared in a microcentrifuge tube 1 as shown in Table 1. Next, transfection solution was prepared in a separate microcentrifuge tube (Tube 2) that contained Lipofectamine™ CRISPR-MAX™ reagent in Opti-MEM™ I reduced serum medium. The concentration was used as described in Table 2 (26.5 µl reaction volume) as per manufacture protocol. The transfection solution

(Tube 2) was directly added to RNPs mix (Tube 1) and mixed well by pipetting up and down. The mix (~50 µl) was next incubated for 10 min at room temperature followed by addition on freshly trypsinized cells in growing phase 0.42–1.2 × $10^5$ cells in 500 µl of the growth medium. The mix and cells with 500 µl growth media were distributed into two wells. Media was replaced after 24 h of incubation, and cells were allowed to grow for 2–3 days. The cells on the first plate were lysed and processed to analyze editing efficiency. The cells on the second plate were cultured for use in assays, banking, and/or single-cell cloning.

### huDR5, Chi-DR5, Chi-G4S-DR5 stable 4T1, and MC38 stable line generation

Transfection of various DR5 constructs (huDR5, Chi-DR5, Chi-G4S-DR5) into the different cell lines (4T1, MC38, ID8) was achieved by jetOPTIMUS DNA transfection reagent for recombinant DR5 cloned in pcDNA3.1 vector. In brief, 60–70% confluent cells were grown in 10cm culture dish. Mixing 10 µg of plasmid DNA and 10 µl of transfection reagent into 1 ml of jetOPTIMUS buffer made transfection solution. After incubating for 10 min at room temperature, the transfection mix was added on the cells. The cells were further allowed to grow for 24 h and then selected using 2 mg/ml of G418. In detail, 10 µg DNA was diluted into 1,000 µl jetOPTIMUS buffer and vortexed. This was followed by addition of 10 µl jetOPTIMUS into the DNA solution (ratio 1:1 corresponding to µg DNA: µl reagent) and vortexed and spun down briefly. Mixture was incubated for 10 min at room temperature. Next transfection mix was added drop wise onto the cells in serum containing medium and distribute evenly. Plates were incubated at 37°C for 24 h. Next day transfection medium was replaced with by cell growth medium and cells were allowed to grow for another day before starting G418

**Table 1. Reagent details for ribonucleoprotein (RNP) preparation**

| RNP Preparation (Tube 1) | | |
|---|---|---|
| **Component** | **Molarity** | **Volume (per reaction)** |
| Opti-MEM™ I Reduced Serum Medium | – | 25 µl |
| sgRNA | 3 µM (pmol/µl) | 1.3 µl (3.9 pmol) |
| Cas9 | 3 µM (pmol/µl) | 1 µl (3 pmol) |
| Lipofectamine™ Cas9 Plus Reagent | – | 1 µl |
| Total volume | – | 28.3 µl |

**Table 2. Reagent details for preparing CRISPR-KO transfection solution**

| Transfection Solution (Tube 2) | |
|---|---|
| **Reagent** | **Volume (per reaction)** |
| Opti-MEM™ I Reduced Serum Medium | 25 µl |
| Lipofectamine™ CRISPRMAX™ Transfection Reagent | 1.5 µl |
| Total volume | 26.5 µl |

2 mg/ml selection. Media was changed every day and reliable GFP signal was evident 72 h of transfection.

### Lentiviral preparation and transduction

Lentiviral packaging and delivery was executed by using the technology from system Biosciences (see Chimeric DR5 cloning section above) and method was very similar to described here (Wollebo *et al*, 2013). Briefly, lentivirus was prepared by transfecting 293T cells in T75 flask with transfer vector (6 μg) and packaging vectors (3 μg each) in the ratio of 2:1:1:1 using 30 μg of PEI. The virus containing culture medium was collected 48 and 72 h after transfection, cleared by filtration (0.45 μm Millipore, Bedford, MA) and concentrated by 20% PEG 6000. After centrifugation at 3000g for 30 min, the pellet was resuspended in $1/10^{th}$ of the initial volume in phosphate-buffered saline (PBS)/0.1% bovine serum albumin (BSA), stored at −70°C. For transduction, the 60–70% confluent cells were plated in 10 cm plate and 5 ml virus along with 5 μg/ml polybrene was added. Transduction medium was replaced with growth medium after 12 h and allowed the cells to grow for another 24 h. The transduced DR5-positive cells were selected using 2.5 μg/ml puromycin. In details, HEK 293T cells were cultured in high glucose-containing Dulbecco's modified Eagle's medium (DMEM; Corning) supplemented with 10% fetal bovine serum (FBS, Corning) at 37°C at 5% $CO_2$. For transfection, the cells were seeded a day before at a density of 70–80% in 10 cm culture dish. Transfection mix was prepared as following: Transfer plasmid with gene of interest (6 μg), pVSVG plasmid (3 μg), pREV (3 μg), pRRE (3 μg), Opti-MEM media (500 μl) and PEI (30 μg). Transfection mixture was vortex mixed and quick centrifuged and incubated in room temperature for 10 min. Transfection mixture was then added gently on the cells through the wall and mix by tilting the plate. Transfected cells were incubated at 37°C for 12–16 h and medium was replaced with 10 ml of fresh growth medium. Virus containing culture media was collected after 48 and 72 h of incubation. The floating cell debris were separated by quick spin at 1,000 rpm for 5 min and then virus containing culture medium was filtered through 0.45 μm Millipore, Bedford, MA. Next virus was concentrated by using PEGylation in the following ratio: Virus suspension (40 ml), 50% PEG 6000 (10 ml) and 5 M NaCl (1 ml). PEGylated solution was mixed and incubated at 4°C overnight on gentle rocker. The precipitated virus particles were centrifuged at 4,000 rpm for 30 min and resuspended in 4 ml of culture medium. To transduce, the tumors cells were seeded a day before at a density of 60–70% in 10 cm culture plate. Transduction solution was prepared by mixing 2 ml of virus, 5 μg/ml of polybrene and 1× HEPES buffer and gently added on the cells with 8 ml of culture medium. Cells were allowed to grow at 37°C for additional 12 h and then medium was replaced with growth medium. The 2.5 μg/ml puromycin selection was performed after 48 h of transduction and medium was replaced each day with intermediate PBS washing to avoid the accumulation of dying cell debris. Stable cells appeared within a week.

### Nude and syngeneic tumor xenograft studies

All animal procedures were conducted under the accordance of University of Virginia Institutional Animal Care and Use Committee (IACUC) approved protocol and conform to the relevant regulatory standards. Details of mouse strains, age and sex used are provided above. Briefly, 6- to 8-week-old (age), 20–25-g (weight),

both male and female (sex) mice were used for tumor xenografts generation, *in vivo* efficacy studies, imaging studies, TIL isolation studies as described in the text. Following mouse strains were used: C57BL/6 (Jackson laboratories), BALB/c (Jackson laboratories), immunodeficient BALB/c-derived athymic Nude $Foxn1^{nu}$/$Foxn1^+$ (envigo), and NOD.Cg-$Prkdc^{scid}$ $Il2rg^{tm1Wjl}$/SzJ also called as NSG mice. Various different solid cancer cell lines were used for tumor nude xenograft studies as described in text. Weight- and age- (6–8 weeks old)-matched mice were injected subcutaneously (SC) in their right flank with indicated cell lines in Matrigel. Different cell number was injected as some cells were highly effective and some required higher density during xenografts. Tumor cells were mixed 100 μl volume with Matrigel. For anti-tumor efficacy studies, mouse bearing ∼100 mm³ tumors weight-matched animals were randomly assigned into groups and injected (either 25 μg or indicated different dose) either intraperitoneally or intravenously (as indicated in figure legends) three times per week with indicated antibodies in text and figure legends. Tumors were measured in two dimensions using a caliper as described previously (Wilson *et al*, 2011; Graves *et al*, 2014). Tumor volume was calculated using the formula: $V = 0.5a × b^2$, where a and b are the long and the short diameters of the tumor, respectively ($n = 4$–6 animals were used for each therapeutic antibody injection). The *P* values are determined by two-tailed paired Wilcoxon–Mann–Whitney test. For syngeneic and surrogate tumor xenograft studies, most experiments were carried out using TNBC 4T1 cell lines. MC38, a colon cancer line, forms tumor in C57BL/6, while 4T1 was only effective in forming tumors in BALB/c immune-sufficient mouse strain. Similar to nude xenograft studies, 6- to 8-week-old female littermate of matched size and weight BALB/c mice were injected subcutaneously (SC) in their right flank with $0.5 × 10^6$ 4T1 cells lines in Matrigel. 4T1 cells very highly consistently in forming tumors (19 out of 20) within ∼2–3 weeks as described (Takeda *et al*, 2008). For tumor regression studies and TIL isolation studies, mouse bearing ∼100 mm³ tumors were (after matching tumor size, number as indicated in legends) randomly assigned into groups and injected with therapeutic antibodies as indicated (50–100 μg dose) intraperitoneally two-three times per week (generally six doses or as indicated in figure legends). Most *in vivo* experiments were repeated three times or as indicated in figure legends. For surrogate efficacy studies, MD5-1, MD5-1 + avelumab and IgG1 control were engineered with IgG1 KO-Fc and S267E mutations. Avelumab was engineered as IgG4-Fc. Tumors were measured two-three times a week and volumes were calculated as the product of three orthogonal diameters similar to nude animal studies as described in previous section. The *P* values are determined by two-tailed paired Wilcoxon–Mann–Whitney test. For biochemical analysis, signal cell isolation, or TIL of tumors, mice were euthanized after indicated antibody treatment followed by tumor extraction (Wilson *et al*, 2011; Li & Ravetch, 2012).

### Quantitation, statistical analysis details

Data unless indicated otherwise are presented SD. In general, when technical replicates were shown for *in vitro* experiments, Student's *t*-test was used for statistical analysis and the same experiment was at least repeated once with similar trend observed. When data from multiple experiments were merged into one figure, statistical significance was determined by either unpaired *t*-test, paired *t*-test, or

## The paper explained

### Problem

Targeting epithelial cell-enriched death receptor-5 (DR5) is a promising therapeutic strategy for solid cancers. However, all clinically tested DR5 agonist antibodies have proved unsuccessful in phase-II trials of solid cancers. As harnessing the immune system has emerged as a powerful tool for oncologic therapeutics, a major issue remains before realizing the true clinical potential of DR5 antibodies: What if undiscovered immune evasion mechanisms counterbalance the anti-tumor activity and could potentially have contributed to the clinical failure for DR5 antibodies?

### Results

We have discovered unexpected findings of programmed death ligand-1 (PD-L1) immune checkpoint receptor activation by DR5 antibodies. We show that DR5 agonist antibody-activated caspase-8 and Rho-associated kinase-1 (ROCK1) signaling regulates PD-L1 stabilization and tumor cell surface mobilization. Further, apoptotic cell-derived extracellular vesicles (ApoEVs) help transfer stabilized PD-L1 pool from DR5-sensitive cells to resistant cells in a mixed heterogeneous tumor, which orchestrates immune-suppressive tumor microenvironment. Co-targeting of either ROCK1 or PD-L1 pathway along with DR5 improves anti-tumor function.

### Impact

Decades of research focused on the deregulation of intrinsic and extrinsic cell death have defined various genetic and non-genetic apoptotic variable factors contributing to acquired resistance in clinical settings. This work defines the comprehensive yet complex interplay of cell death and immune dysfunction and provides the groundwork for future intervention therapies. Considering ROCK-1 function in actin polymerization regulation, endosomal recycling, and membrane blebbing, its direct association with PD-L1 and CMTM6 complex could serve as an additional therapeutic target with already successful PD-L1 therapies to avoid immune tolerance in cytotoxic tumors. If superior cell death could be maintained by keeping immune suppression in check, a potential new therapeutic avenue will open doors to give a second lease of life to clinically tested DR5 agonist antibodies to enhance tumor immunity and overpower clinical efficacy.

Wilcoxon–Mann–Whitney test as indicated under Data information at the end of figure legends using GraphPad Prism 5.0 software. For animal studies, tumor-bearing animals prior to treatment of different antibodies (which are being compared to each other) were randomly selected. To minimize bias if there were animals that had initial smaller tumor size, they were equally distributed to each treatment group. There was no subjective bias in animal selection. For experiments where treatment results in significant differences in tumor size (between treatment groups), the follow-up analysis was carried out by tumor size match to avoid variations before statistical comparison. Tumor growth curves are displayed as mean ± SEM. For all the statistical experiments, $P$ values, $P < 0.05$ (*), $P < 0.005$ (**), $P < 0.0001$ (***), and $P < 0.00005$ (****) were considered statistically different or specific $P$ values indicated otherwise or "ns" indicates non-significant.

## Data availability

This study includes no data deposited in external repositories. Requests for resources and reagents generated during this study should be directed to and will be fulfilled by Lead contact: Jogender Tushir-Singh (jogi@virginia.edu). We are happy to share various reagents and antibodies (their sequences) used in this study as long as the requests does not infringe on intellectual proprietary and licensing.

**Expanded View** for this article is available online.

## Acknowledgements

We are thankful to University of Virginia Cancer Center Core Imaging Facility, Core Flow Cytometry Facility, IHC core, Biomolecular Analysis Facility, Advanced Microscopy Facility, and the Core Vivarium Facility for Assistance. JT-S is an early career investigator of Ovarian Cancer Academy, U.S. Department of Defense (OCA-DoD). SB is Hartwell Foundation Investigator. This work was supported by NCI/NIH grant (R01CA233752) to JT-S, U.S. DoD Breast Cancer Research Program (BCRP) breakthrough level-1 award to JT-S (BC17097) and SB (BC170197P1), U.S. DoD Ovarian Cancer Research Program (OCRP) funding award (OC180412) to JT-S and P30CA044579 Cancer Center Support Grant to the University of Virginia.

## Author contributions

Conceptualization: TM, GNS, SB, JT-S; Methodology: TM, GS, SB, JT-S; Technical support for methodology: EL, RGTT, MB, DT, RM, KU, NSR, MSS; Analysis: TM, GNS, SB, ES, JT-S, Resources: NSR, JCH, PDB, MSS; Writing Original Draft: SB, JT-S; Supervision: JT-S.

## Conflict of interest

The authors declare that they have no conflict of interest. Co-targeting of DR5 + PD-L1 and DR5 + ROCK1 for improved solid cancer immunotherapy is part of a provisional patent here at the University of Virginia Licensing and Venture Group.

## For more information

https://www.cancer.gov/publications/dictionaries/cancer-drug/def/lexa tumumab

https://www.cancer.gov/publications/dictionaries/cancer-drug/def/tigatuzumab

https://www.cancer.gov/publications/dictionaries/cancer-drug/def/cona tumumab

https://www.cancer.gov/publications/dictionaries/cancer-drug/def/apomab

https://clinicaltrials.gov/ct2/show/NCT00428272

https://clinicaltrials.gov/ct2/show/NCT01307891

https://clinicaltrials.gov/ct2/show/NCT01327612

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
