## [Review Process File · EMBO Molecular Medicine]

Unexpected PD-L1 Immune Evasion mechanism in TNBC, Ovarian and other Solid Tumors by DR5 Agonists

Tanmoy Mondal, Gururaj Shivange, Rachisan Tihagam, Evan Lyerly, Michael Battista, Priya Talwar, Roxanna Mosavian, Karol Urbanek, Narmeen Rashid, Joshua Harrell, Paula Bos, Edward Stelow, M. Sharon Stack, Sanchita Bhatnagar, and Jogender Tushir-Singh

DOI: 10.15252/emmm.202012716

Corresponding author(s): Jogender Tushir-Singh (jogi@virginia.edu) , Sanchita Bhatnagar (sb5fk@virginia.edu)

Review Timeline:	Submission Date:	13th May 20
	Editorial Decision:	3rd Jun 20
	Revision Received:	26th Oct 20
	Editorial Decision:	21st Oct 20
	Editorial Decision:	23rd Nov 20
	Revision Received:	3rd Dec 20
	Accepted:	7th Dec 20

Editor: Zeljko Durdevic

Transaction Report:

3rd Jun 2020

Dear Dr. Tushir-Singh,

Thank you for the submission of your manuscript to EMBO Molecular Medicine. We have now heard back from the three referees who agreed to evaluate your manuscript. As you will see from the reports below, the referees acknowledge the interest of the study. However, they raise some concerns that should be addressed in a major revision of the present manuscript. Particular attention should be given to the validation of the findings in a clinically more relevant in vivo model. Addressing the reviewers' concerns in full will be necessary for further considering the manuscript in our journal. Please make sure to pay special attention to the grammar and syntax and I would recommend running the article by a native English speaker.

Acceptance of the manuscript will entail a second round of review. Please note that EMBO Molecular Medicine encourages a single round of revision only and therefore, acceptance or rejection of the manuscript will depend on the completeness of your responses included in the next, final version of the manuscript. For this reason, and to save you from any frustrations in the end, I would strongly advise against returning an incomplete revision.

We realize that the current situation is exceptional on the account of the COVID-19/SARS-CoV-2 pandemic. Therefore, please let us know if you need more than three months to revise the manuscript.

I look forward to receiving your revised manuscript.

***** Reviewer's comments *****

Referee #1 (Remarks for Author):

- In the introduction, it is mentioned that immunotherapy has been disappointing in lung cancer and triple negative breast cancer. This should be mitigated, especially in light of the latest results presented at the AACR 2020 virtual meeting on triple negative breast cancer (Puztai L AACR 2020) and the established prolonged survival of patients treated with immunotherapy in non small cell lung cancer.

- In Figure 1 E and G, it would be important to show a picture of Facs. Does anti-DR5 have an effect on the MFI of PD-L1 or the % of cells expressing PD-L1
- What is the duration of the effect of ApoEV on the transfer of PD-L1 expression to other initially negative cells ?
- It is often stated that inhibition of PD-L1 expression increases the recruitment of CD25-expressing T-CD4 lymphocytes. It should be investigated whether these cells are effector or regulatory T cells (Foxp3+).
- Since it is said that MD5-1 antibodies requires Fc cross-linking to activate cell death, does the biospecific anti-DR5-anti-PD-L1 antibody bind to the Fc. In addition, avelumab initially work at least in part via ADCC.

Referee #2 (Remarks for Author):

In this study, the authors aimed to investigate the immune-mediated mechanisms that contributed to limited efficacy seen with DR5 agonist antibodies in solid tumor. The authors demonstrate that DR5 agonist antibodies stabilized PD-L1 in different solid tumors by activating the DISC complex (caspase-8 activation). The authors found that ROCK1 was a key regulator of PD-L1 surface mobilization and that the combination of ROCK1 inhibitors with anti-PD-L1 antibodies and DR5 agonists could improve anti-tumor efficacy. The study is interesting, significant and provides new insights about potential immune suppression by DR5 agonists, which might be relevant to their failures. However, there are several concerns regarding the results that need to be addressed to strengthen the conclusions.

Specific comments:

1. In the "Introduction" section, the authors link the limited immune cell penetration in solid cancer with PARA and DR5/TRAIL-R2-activating antibodies. It is not clearly apparent why. Similarly, the choice of studying PD-L1 in the context of co-targeting DR5 and ROCK1 appears arbitrary/biased. More background and rationale for this choice need to be provided in the introduction part.
2. In the "Results" section, it is not clear how the authors determined that "PDL1 stabilization requires function of an effector which is potentially is regulated by DISC caspase activation" and that "loss of proteasome function and DISC activation serves same purpose"?
1. In Results (page 3), the authors conclude that blockade of caspase-8 containing death inducing signaling complex (DISC), downstream of DR5 inhibited PD-L1 stabilization (Figure 1C). But the data are from only one cell line (OVCAR-3 cells). Thus, this result as well as many of the results discussed below need to be confirmed using other cancer cell lines.
2. In Results (page 4), it is indicated that "When tested, both transcription and translation inhibitors did not reversed PD-L1 stability generated by DR5 agonists (Figure S5A). "But the figure only shows data from the KMTR+Lexa treated group; what is the impact of pre-treatment with cycloheximide in KMTR+Lexa group? Moreover, the rationale for the experiments presented in Figures 2 and S5, and the interpretation of those data are difficult to understand. Please clarify.
3. In Results (page 5), the authors claim that they observed "accumulation of activated caspase-8" and no activation of ROCK1. These statements are not consistent with the western blot results shown Figure 3G-I.
4. For the animal experiments, only tumor growth data were provided to show anti-tumor efficacy but not any survival data. Can the combination of ROCK1 inhibitors/anti-PD-L1 antibodies and DR5 agonists increase survival in these models?
5. The authors have shown synergistic efficacy of the combination of ROCK1 inhibitors/anti-PD-L1 antibodies and DR5 agonists. What about the triple combination of these drugs?

6. The authors used the combination of different DR5 agonist antibodies in some experiments. What was the rationale for the use of such combination?
7. Statistical analyses were reported for some of the experiments. The authors should include these along with the details on the methods used.
8. The suppression of the activity of incoming cytotoxic T-cells in tumor microenvironment by surface stabilized PD-L1 on dying DR5 sensitive tumor cells is intriguing. But how can the authors exclude other mechanisms, such as increased cytokine secretion in the tumor microenvironment?

Minor points:

1. In "Introduction" section, some statements are not accurate. Immunotherapy has now multiple indications in lung and breast cancers.
2. Some western blot images are of suboptimal quality. For example, Figure 1B, Casp-3 bands; Figure 1M, GAPDH bands; Figure 2L, DR5 bands. These images need to be replaced.
3. Figure 2M and N, the color representing DR5-KO cells and DR5-WT cells should be consistent.
4. Many typos need to be corrected. For example, in the "Introduction" section "Although these strategies improve DR5 activation, it is well establish...", paragraph 2, page 6, "lane 5th lane"; paragraph 2, page 7, "Figure M, P"; Figure S1, A and B are missing.
5. In Figure 1F, authors induced transient epithelial to mesenchymal transitions, while the Vimentin protein level at 0.1X Times seems higher than that at 0.5X Times.
6. In the "Results" section, it is stated that: "TNF- α stabilized both PD-L1 and CSN5 in ovarian and TNBC tumor cells without activating caspases (Figure 2E-G)." Figure 2E-G includes data with only one cell line.
7. The research involves multiple agonists or inhibitors for different molecules such as DR5, PD-L1, etc. Although their biological functions have been summarized in the supplementary table, it is still necessary to clearly explain it in the corresponding figures.
8. More detailed and specific information need to be provided in Table S2.
9. In Results (page 7), it said that "When tested in 4T1 TNBC syngeneic tumor models, Avelu-MD5-1 completely regressed tumors (Figure M, P, S9E)." The callout should be to Figures 6M, 6P, S9E).
10. The manuscript should be carefully edited for clarity and grammar.

Referee #3 (Remarks for Author):

This manuscript aims to investigate the clinical failure of DR5 agonists as anti-cancer agents. They find that PD-L1 protein and surface expression increase in DR5 agonist-treated cell lines and xenografts, and that this is dependent on signaling downstream of DR5 as the increase is blocked by caspase 8 (but not 3) inhibition. Resistance to DR5 reverses the phenomenon. They show that DR5 agonist-treated tumor cells reduce activity of CD3-stimulated Jurkat T cells in co-culture, and that increased PD-L1 following DR5 agonist treatment is via enhanced stability, by proteasomal inhibition (S5a) not transcriptional or translational routes. They then suggest that cells dying in response to DR5 agonists release their PD-L1 in EVs which can transfer to neighboring insensitive cells, allowing the insensitive cells to inhibit T cell activity in co-culture.

Mechanistically, the authors find that ROCK1 becomes activated in DR5 agonist treated cells and since it isn't inhibited by autocatalytic blockade of caspase 3, it may be directly activated by DR5/caspase 8, although there is no evidence for this. Consequently, ROCK inhibition could downregulate surface PD-L1 without affecting DR5 agonist-induced cell death. ROCK1's ability to enhance surface PD-L1 may be due to complex formation with PD-L1 and CMTM6.

Therapeutically, the authors show that combining ROCK1 inhibitor and anti-DR5 significantly delays xenograft growth in vivo and suggest that ROCK1 blockade in presence of DR5 agonists, conditions and help mobilize activated T-cells in TME by inhibiting PD-L1 surface mobilization. The authors also made a chimeric human/mouse DR5 fusion to test clinical DR5 agonists and found that PD-L1 was still elevated following agonist treatment, whilst ROCK inhibition in combination could reduce tumor growth and allow T cell infiltration, as could PD-1 blockade. CD8 T cell depletion abrogated the efficacy of the combination treatment. Finally, the authors engineered a bi-specific antibody against DR5 and PD-L1 which improved efficacy (although they haven't performed the side by side comparison).

There are some really interesting findings here, but the paper is quite challenging to follow and would benefit from more detail in the text rather than having to find it in supplemental figure legends. Grammatically the manuscript also needs some work. The manuscript also suffers from over-interpretation and hyperbolic claims re DR5 agonists (as an example, in the abstract the authors state that solid tumor enriched DR5 targeting antibodies remain a crucial therapeutic strategy, which is clearly overstatement). N numbers and stats are also missing in many places. More specific criticisms/comments are outlined below:

1. It would really help the paper if the authors could show that this phenomenon is actually occurring in the failed trials of DR5 agonists. They could use IHC or ISH for PD-L1 in patient samples from trials if they could access these?

Failing that, the authors need to test their strategy in a more clinically relevant GEMM model.

2. There are some problems with their overall concept - T cells are excluded from many solid tumors, not just exhausted. So how can ICP blockade or ROCK inhibition correct this, unless DR5 agonists somehow increase T cell infiltration?

3. In fig1a the westerns for PD-L1 are not terribly convincing - quantification of replicates should be included. A PARP cleavage western should also be included in fig 1M. Where are the error bars and stats for fig 1E and K? Presumably the authors performed the experiment more than once? And how are there stats for 1G, but no error bars?

4. There is a huge variation in drug concentrations used in the T cell co-culture assays (e.g. from 5nM to 1uM for KMTR2). How many off-target effects may be driven by 200-fold increases in drug concentration? Can caspase inhibition still rescue T cell activation when using 1uM concentrations? Also, the authors can't say that these experiments confirm PD1-PD-L1 interaction. They should include PD-1 blocking antibody and determine whether the DR5 agonists no longer drive loss of T cell activity.

5. In order to confirm that the EV transfer of PD-L1 is a DR5-mediated phenomenon, they should include non-DR5 targeting therapy here.

6. The authors state that ROCK1 cleavage was consistently observed prior to full caspase-3 activation. They cannot make that statement based on westerns of varying quality and intensity. They suggest that these data support a direct role of caspase-8 (and DISC) in ROCK1 cleavage, which they do not have evidence for. They should at least use a caspase 8 inhibitor to verify this. They also conclude that ROCK1 activation is a key regulatory second step in order to surface mobilize the internally stabilized PD-L1 on cell-surface - unless they conduct some live cell imaging experiments, they can't conclude anything about ROCK1's involvement in mobilization.

7. When testing therapeutic strategies in vivo, the authors mostly show pictures of tumor size but don't suggest how long treatment was for - were these tumors excised at end point? At a specific timepoint?
8. In xenograft studies, the tumor size will affect TIL infiltration. The authors need to show PD-L1 expression, T cell penetrance and cleaved caspase 3 in vivo (and in situ e.g. by IHC, not from homogenized tumor), and preferably control for tumor size - either IHC to see where T cells are, or perform the T cell FACS in size matched tumors.
9. The authors make the unusual claim that GSK269 reduced tumors growths only at high >10mg/kg dose, while 1mg/kg dose in their cellular assay was effective to inhibit PD-L1 mobilization. How exactly do they calculate mg/kg in a cellular assay?
10. When examining the TILs, following treatment with the bi-specific antibody the authors claim that, as expected, MD5-1 treatment inhibited TIL mobilization into tumors. That is not what figure 6N shows. Can the authors explain this?

Minor

1. S2B axes don't make any sense. Neither does the legend: "(B) Same as B in colo-205 cells"
2. I wouldn't label DR5-sensitive cancer cells as 'WT'
3. In fig S3 the legend states 4-6 doses, figure suggests 2-3. Which is right?
4. Why is the ROCK western in fig S6A so different to the rest?
5. Fig S9 is labelled S10
6. In the clinical agonist the authors fail to show PD-L1 expression in DR5 vs DR5/ROCK inhibitor treated tumors. I'm sure they have this data.

Please find below the point-by point response to every single comment raised by reviewers. We have addressed almost all the issues raised (in the revised manuscript).

In response to Reviewer-1: We have addressed his/her concern **in full** as follows:

1) In the introduction, it is mentioned that immunotherapy has been disappointing in lung cancer and triple negative breast cancer. This should be mitigated, especially in light of the latest results presented at the AACR 2020 virtual meeting on triple negative breast cancer (Puztai L AACR 2020) and the established prolonged survival of patients treated with immunotherapy in non small cell lung cancer.

Response: We agree with the Reviewer's concern. On reviewer's suggestion, we have reworded the introduction section and have added a reference supporting the successful approval of atezolizumab (impassion trial) immunotherapy against TNBC. Please see the 3rd paragraph of introduction.

2) In Figure 1 E and G, it would be important to show a picture of Facs. Does anti-DR5 have an effect on the MFI of PD-L1 or the % of cells expressing PD-L1

Response: We agree with the Reviewer. To accommodate reviewer's request, we have added representative FACS in the revised manuscript (Please see Fig 1G).

Yes. DR5 agonist have effect on both: a) MFI of PD-L1 and, b) % cells expressing PD-L1. On reviewer's suggestion, new data (with MFI analysis) has been added in Appendix Fig S2B.

3) What is the duration of the effect of ApoEV on the transfer of PD-L1 expression to other initially negative cells?

Response: We strongly agree with the reviewer's comments and experimental suggestions. In response, we have carried out PD-L1 transfer kinetics (from ApoEVs) and new data sets have been added in Fig 3G, Appendix Fig S6, and Fig EV3A. In terms of duration kinetics, using ApoEVs derived from two different cell lines, we consistently observed increased PD-L1 surface stability in DR5 KO lines 24-48 hrs post addition of ApoEVs.

4) It is often stated that inhibition of PD-L1 expression increases the recruitment of CD25-expressing T-CD4 lymphocytes. It should be investigated whether these cells are effector or regulatory T cells (Foxp3+).

Response: We strongly agree with the reviewer's comments and appreciate experimental suggestions. In response, we have carried out FoxP3 (a marker of T-regs) IHC studies and western blots from size matched tumors after various treatments. The new data sets have been added in Fig 6D, Fig 6K, Fig EV5 and Fig EV4E. We did not observe enrichment of FoxP3⁺ regulatory T-cells in tumors. To further strengthen our findings, we have also added many other new data sets to manuscript:

a) $cd8^+CD45^+$ T-cell enrichment data (flow cytometry data) in Fig 6A-C and Appendix Fig S10,
b) High tumor granzyme-b (an indicator of cytotoxic t-cell function) activity western blotting data after treatment of DR5 antibody +ROCK1i and DR5 antibody +avelumab in Fig 6D, Fig 6K, Fig EV4E.

c) High IFN- γ expression data (flow cytometry) in Fig 6L, M and Appendix Fig S11A-C from size matched tumors after various indicated treatments. These additional sets of data collectively support our hypothesis of enriched cytotoxic T-cells rather than regulatory T-cells (Tregs) after combinatorial treatments. However, we do not rule out the possibility of Tregs playing a negative role in other tumor models or other therapeutic interventions (included in discussion).

5) Since it is said that MD5-1 antibodies requires Fc cross-linking to activate cell death, does the biospecific anti-DR5-anti-PD-L1 antibody bind to the Fc. In addition, avelumab initially work at least in part via ADCC.

Response: We appreciate reviewer's comment and Thank you for pointing this out. We would like to clarify that neither monospecific nor bispecific antibody bind to FcYR11a, as they carry L234A L235A (LALA) mutation. Monospecific MD5-1, lexatumumab KMTR2 and all DR5 agonist were engineered with LALA mutations to avoid interference from ADCC and complement function.

These details (including a reference citation) have been added in the first paragraph of result section under: "PD-L1 stabilization by DR5 agonist antibodies in solid tumors".

In response to Reviewer-2: We have addressed his/her concern **in full** as follows:

Specific comments:

1. In the "Introduction" section, the authors link the limited immune cell penetration in solid cancer with PARA and DR5/TRAIL-R2-activating antibodies. It is not clearly apparent why. Similarly, the choice of studying PD-L1 in the context of co-targeting DR5 and ROCK1 appears arbitrary/biased. More background and rationale for this choice need to be provided in the introduction part.

Response: We agree and appreciate Reviewer's suggestion. In response, we have updated the introduction background and rationale segment to justify PARA and DR5 agonist targeting strategy an alternative therapy to breakdown tumors to support increased immune cell penetration and solid tumors. Our new sets of data also support enhanced T-cell (immune) infiltration in tumors upon DR5 agonist treatments (Fig 6A-C, Fig 6D, Fig 6K, Fig 6L-M, Fig EV4E). We have also added CD8, CD4, Fox3p etc. IHC data (Fig EV5).

We have also updated the introduction to provide background and rationale of studying PD-L1 and DR5 co-targeting as per reviewer's request (Please see 3rd paragraph of introduction section).

2. In the "Results" section, it is not clear how the authors determined that "PDL1 stabilization requires function of an effector which is potentially is regulated by DISC caspase activation" and that "loss of proteasome function and DISC activation serves same purpose"?

Response: We appreciate Reviewer's comment. We have updated the text in result under "CSN5 is not required for PD-L1 upregulation by DR5 agonist antibodies" section. The new language has been simplified which now states as follows: 1) "Strikingly, proteasome inhibition (for a longer period) increased overall PD-L1 in tumor cells lysates (Fig 2K) and DR5 agonist plus MG132 co-treatment did not additionally stabilize PD-L1 on cell surface or in total lysates (Fig 2I-J, Fig EV2G, Appendix Fig S5A). These results indicated that proteasome inhibition (by MG132) was a linear and downstream event of DR5 agonist signaling. As DR5 agonist antibodies functions via direct caspase-8 activation in DISC and given the reports of caspase mediated proteasome inactivation(Cohen, 2005), we next explored the possibility that PD-L1 stabilization is a byproduct of proteasome inactivation".

2) "Collectively these results confirm two different proteasome interference mechanisms of PD-L1 stabilization, where one works by deubiquitinating PD-L1 (Lim et al., 2016), while other works by degrading proteasome subunits"

1. In Results (page 3), the authors conclude that blockade of caspase-8 containing death inducing signaling complex (DISC), downstream of DR5 inhibited PD-L1 stabilization (Figure 1C). But the data are from only one cell line (OVCAR-3 cells). Thus, this result as well as many of the results discussed below need to be confirmed using other cancer cell lines.

Response: We agree with the Reviewer. Upon reviewer request, data from another triple negative breast cancer cell line (MDA-MB-436) has been added along with ovarian cancer cell line (OVCAR-3). The data from two different cell line is present in Fig 1C and Fig EV1A. Similarly, for CSN5 independent PD-L1 stabilization studies, additional data from another cancer line (OVCAR-3) has been added. The data from two different cell line is present in Fig 2G and Fig EV2F.

2. In Results (page 4), it is indicated that "When tested, both transcription and translation inhibitors did not reversed PD-L1 stability generated by DR5 agonists (Figure S5A). "But the figure only shows data from the KMTR+Lexa treated group; what is the impact of pre-treatment with cycloheximide in KMTR+Lexa group? Moreover, the rationale for the experiments presented in Figures 2 and S5, and the interpretation of those data are difficult to understand. Please clarify.

Response: Yes, we agree with the reviewer and appreciate his/her comment. New cycloheximide data has been added in Fig EV2E. As expected cycloheximide pretreatment (2hr and 4hr) did not change PD-L1 stabilization by DR5 agonists. At 8hr cycloheximide pretreatment, we did observe decreased PD-L1 compared to 2 hrs and 4hrs. However, it must be noted that PD-L1 levels (8 hrs cycloheximide) were still higher compared to IgG1 control or non-DR5 agonist treated control (compare lane 4, 5 vs lane 9). Moreover, 8 hrs pre-cycloheximide treatment also reduced overall PD-L1 in IgG1 treated samples suggesting general translation blockade rather PD-L1 specific translational blockade.

As per Reviewer's request we have also simplified the rationale of 26S proteasome regulatory submit S5a and its connection with DR5 agonist and PD-L1 stabilization in result section under "CSN5 is not required for PD-L1 upregulation by DR5 agonist antibodies".

3. In Results (page 5), the authors claim that they observed "accumulation of activated caspase-8" and no activation of ROCK1. These statements are not consistent with the western blot results shown Figure 3G-I.

Response: Yes, we strongly agree with the Reviewer's comment. "Accumulation" has been replaced with "steady activity". We have updated the statement under "Role of ApoEVs in PD-L1 stabilization in heterogeneous tumors" section in the last paragraph, which is now consistent with western blot data. The updated statement read as follows:

With KMTR2 and Z-DEVD co-treatments, we observed steady activity of caspase-8 (as evident by its cleavage), while the activity of prodomain containing cleaved caspase-3 was inhibited, if Z-DEVD was added to the cultures within 40 minutes of DR5 agonist (KMTR2) treatment. The latter is evident by loss of caspase-3 autocatalytic event (lack of accumulation of 17KDa, see lane 4-6 in Fig 3K), loss of PARP cleavage (see lane 4-6 in Fig 3K). The results were also supported by caspase-3 activity assays (Fig 3L) and cell death assays (Fig 3M). Importantly S5a levels were also reduced and PD-L1 was stabilized in lysates despite lack of effective cell death (Fig 3K, lane 2-6). These findings are also suggestive that DR5 agonist antibodies with high extrinsic ability to generate activated DISC-caspase-8 (despite not executing cell-death above tumor clearance threshold) can also potentially stabilize PD-L1. The latter could be a contributing factor in tumor cells having dysregulation of downstream pro-survival proteins such as Bcl-2 (and Bcl-xL etc.) despite optimal DR5 activation.

4. For the animal experiments, only tumor growth data were provided to show anti-tumor efficacy but not any survival data. Can the combination of ROCK1 inhibitors/anti-PD-L1 antibodies and DR5 agonists increase survival in these models?

Response: We agree with the Reviewer's comment. Upon reviewer's request, along with detailed efficacy and survival data of double (DR5 antibody +ROCK1i, DR5 antibody +avelumab) and triple combination (DR5 antibody +ROCK1i +avelumab) has been added. The new data is in Fig5O-Q.

5. The authors have shown synergistic efficacy of the combination of ROCK1 inhibitors/anti-PD-L1 antibodies and DR5 agonists. What about the triple combination of these drugs?

Response: We agree with the Reviewer's comment. Upon reviewer's request, along with detailed efficacy and survival data of double (DR5 antibody +ROCK1i, DR5 antibody +avelumab) and triple combination (DR5 antibody +ROCK1i +avelumab) has been added. The new data is in Fig5O-Q.

6. The authors used the combination of different DR5 agonist antibodies in some experiments. What was the rationale for the use of such combination?

Response: We totally understand reviewer's concern. We would respectfully like to clarify that some experiments, instead of using 100nM one antibody, 50+50nM combinations of two antibodies were used. There are two reasons for that 1) DR5 agonist combinations have shown to increase their activity against partially sensitive cell lines. Please see- doi: 10.1016/j.ccr.2014.04.028, Cancer Cell 2014 Aug 11;26(2):177-89 . Therefore, some cell lines such as MDA-MB-231, A549 which are lesser sensitive to DR5 agonist, we activated DISC-caspase-8 activity by using combination of antibodies to support our hypothesis. 2) The second rationale for these experiments was to show that PD-L1 stabilization is independent of particular antibody binding to particular epitope of DR5 rather because of higher DISC-caspase-8 activity. We also understand the rationale in the context can be confusing so we have removed redundant data generated from dual combinations wherever possible.

7) Statistical analyses were reported for some of the experiments. The authors should include these along with the details on the methods used.

Response: We agree with the Reviewer. Statistical analyses and details have been added in all the figures wherever possible.

8. The suppression of the activity of incoming cytotoxic T-cells in tumor microenvironment by surface stabilized PD-L1 on dying DR5 sensitive tumor cells is intriguing. But how can the authors exclude other mechanisms, such as increased cytokine secretion in the tumor microenvironment?

Response: We agree with the reviewer with cytokines secreted by MDSCs or Tregs could also influence activity of incoming cytotoxic T-cells. As per reviewer's suggestion language has been updated in the discussion section. (Please see the 2nd last paragraph of discussion).

Minor points:

1. In "Introduction" section, some statements are not accurate. Immunotherapy has now multiple indications in lung and breast cancers.

As per reviewer's request, the introduction section has been updated to include the success of immunotherapy and FDA approval in TNBCs (Please see 3rd paragraph of introduction section).

2. Some western blot images are of suboptimal quality. For example, Figure 1B, Casp-3 bands; Figure 1M, GAPDH bands; Figure 2L, DR5 bands. These images need to be replaced.

We agree with reviewer's suggestion. As per reviewer's request many western blots including the Figure 1B, Casp-3 bands; Figure 1M, GAPDH bands; Figure 2L, DR5 bands has been replaced. In addition, other blots which we felt were of suboptimal quality has been replaced.

3. Figure 2M and N, the color representing DR5-KO cells and DR5-WT cells should be consistent.

We completely agree with reviewer's suggestion. As per reviewer's request, we have updated the representative colors for DR5-KO cells and DR5-WT cells.

4. Many typos need to be corrected. For example, in the "Introduction" section "Although these strategies improve DR5 activation, it is well establish...", paragraph 2, page 6, "lane 5th lane"; paragraph 2, page 7, "Figure M, P"; Figure S1, A and B are missing.

We highly appreciate reviewer's through reading and point out these mis-labels. We have updated all of them now.

5. In Figure 1F, authors induced transient epithelial to mesenchymal transitions, while the Vimentin protein level at 0.1X Times seems higher than that at 0.5X Times.

Thanks to reviewer for point it out. We have replaced the Vimentin western blot. The vimentin blot in revised manuscript is Fig EV1C

6. In the "Results" section, it is stated that: "TNF- α stabilized both PD-L1 and CSN5 in ovarian and TNBC tumor cells without activating caspases (Figure 2E-G)." Figure 2E-G includes data with only one cell line.

We totally agree and appreciate reviewer's suggestion. Additional data from another cancer line (OVCAR-3) has been added. The data from two different cell line is present in Fig 2G and Fig EV2F.

7. The research involves multiple agonists or inhibitors for different molecules such as DR5, PD-L1, etc. Although their biological functions have been summarized in the supplementary table, it is still necessary to clearly explain it in the corresponding figures.

We totally agree with reviewer's suggestion. Details of various molecular agonists or inhibitors have been updated now in corresponding figures.

8. More detailed and specific information need to be provided in Table S2.

We totally agree with reviewer's suggestion. Additional and specific information has been updated in Table S2

9. In Results (page7), it said that "When tested in 4T1 TNBC syngeneic tumor models, Avelu-MD5-1 completely regressed tumors (Figure M, P, S9E)." The callout should be to Figures 6M,6P, S9E).

We agree with reviewer's comment. New data has been added and callout has been updated.

10. The manuscript should be carefully edited for clarity and grammar.

We appreciate reviewer's Suggestion. The manuscript has been edited for clarity and grammar.

In response to Reviewer-3: We have addressed his/her concern **in full** as follows:

Major comments

1. It would really help the paper if the authors could show that this phenomenon is actually occurring in the failed trials of DR5 agonists. They could use IHC or ISH for PD-L1 in patient samples from trials if they could access these? Failing that, the authors need to test their strategy in a more clinically relevant GEMM model

Response: We strongly and respectfully agree with the Reviewer and we also agree with his/her suggestion that IHC or ISH for PD-L1 in patient samples from DR5 clinical trials would be a great addition to support the described phenomenon in the paper. In response, before submitting this manuscript, I did reach out to a few pharmaceuticals (Genentech) and clinician investigators at University of Alabama to get access of patient samples for IHC and ISH studies, where failed lexatumumab, tigatuzumab (anti-DR5 agonist) trials were carried out in TNBC patients few years ago. Unfortunately, I have not received any response from them till today. It seems impossible to get access to these samples for multiple reasons including potential signed patient privacy rights prior to trials.

In response to reviewer's concern and to support our finding we have carried out PD-L1 IHC studies using human patient derived tumor xenografts treated with DR5 agonist KMTR2. The new data support positive high staining of PD-L1 in KMTR2 treated tumor as compared to IgG1 treated tumors. The new data has been added in Fig 1L and Appendix Fig S2C.

Regarding the second point of GEM model (GEMM), we also agree with reviewer that it will add additional data to support phenomenon. In response, we have tested DR5 expression in Breast TNBC GEM models MMTV-PyVT with the help of Dr. Paula Bos (contributing author added from Virginia Commonwealth University) who is expert with T-regs and TNBC GEM models. Unfortunately, <10% MMTV-PyVT GEMM tumors cells were positively stained for DR5, while >90% stained for DR5 in syngeneic 4T1 and MC38 tumors (n=3, raw data provided, please see Appendix Fig S11D). Thus, based on expression MMTV-PyVT GEM model does not represent the right animal population to test the described approach. Furthermore, based on recently published studies, other TNBC GEM models also (such as MMTV-neu in addition to MMTV-PyVT) express undetectably low PD-L1 (<5%) on tumor cells (DOI: [10.1126/scitranslmed.aal4922](https://doi.org/10.1126/scitranslmed.aal4922) , see Fig 3) and do not respond well to PD-L1 therapies. Another GEM breast P53 high mutational burden model (Trp53^{tm1Brd} Brca1^{tm1Aash}) although express around 25% PD-L1 on cell surface, is refractory to PD-1/PD-L1 and anti-CTLA immune checkpoint therapies (See DOI: [10.1126/scitranslmed.aal4922](https://doi.org/10.1126/scitranslmed.aal4922) , Fig 3 d, e). These GEMM tumors only responded when combination of two immunotherapies (anti-PD-L1 and anti-CTLA4) were given alongside of cisplatin. Thus, I would respectfully point out to Reviewer our reservations to test the effect of targeting PD-L1 stabilization (our findings by DR5-ROCK1 mechanism) in non-optimal tumor models that do not respond to PD-L1 therapies.

Unfortunately, right immune competent tumor model is a challenge in DR5 field, which is why we made transgenic surrogate models to test human clinical DR5 agonists in immune competent models (Fig 5A). This is also first ever report of testing human DR5 clinical antibodies in this setting using transgenic surrogate grafts. Importantly, we have tested our findings not only in transgenic surrogate grafts (Chi-G4S-DR5 expressing 4T1 tumors and Chi-G4S-DR5 expressing MC38 tumors) but also in syngeneic grafts (4T1 tumors and MC38 tumors). In addition, we have used more than one clinical DR5 agonist antibody (such as lexatumumab, KMTR2, tigatuzumab, AMG655 etc) to support our detailed findings.

In revised manuscript we have also added additional survival data which although was not requested by reviewer (See Fig 5O, 5Q), but directly supports the hypothesize.

2. There are some problems with their overall concept - T cells are excluded from many solid tumors, not just exhausted. So how can ICP blockade or ROCK inhibition correct this, unless DR5 agonists somehow increase T cell infiltration?

Response: We respectfully agree with the reviewer comment. In response significant new data has been provided from sized matched tumors, which collectively supports increase immune T-cells infiltration in solid tumors by DR5 agonist.

Please see Fig 6A-C, Fig 6D, Fig 6K, Fig 6L-M, Fig EV5, Fig EV4E and Appendix Fig S10, Appendix S11A-C).

The data in Fig 6A-C and Fig 6L-M has been generated with n=6-20 tumor bearing animals and isolated size match tumors (3 Independent experiments).

The data in Fig 6D, Fig 6K and Fig EV4E supports high granzyme-b activity (an indicator of cytotoxic T-cell function) in tumors consistent with enriched CD8 signal.

The IHC Data in Fig EV5 also supports data in Fig 6A-C, Fig 6D, Fig 6K and Fig EV4E.

The Data in Fig 6L-M and Appendix Fig S11A-C support high IFN- γ expression (flow cytometry) from size matched tumors after various indicated treatments. These newly added data sets strengthen DR5 agonist role in tumor breakdown to potentially increase immune infiltration.

Regarding the second immune infiltration point raised, we would respectfully like to point out to Reviewer that multiple reports have described not only exhaustion but also limited infiltration and exclusion of T-cells in solid tumor microenvironment due to its sturdiness (please see T-cell exclusion section in this doi: [10.1016/j.ccell.2017.02.008](https://doi.org/10.1016/j.ccell.2017.02.008)). Please also see the following review article: DOI: [10.1158/2159-8290.CD-13-0985](https://doi.org/10.1158/2159-8290.CD-13-0985). Importantly, it is also well established that reduction of tumor load, by either by apoptosis inducing cytotoxic drugs mediated breakdown or by surgical tumor breakdown (debulking) always increase immune and T-cell infiltration (DOI: [10.1038/s41598-019-52913-z](https://doi.org/10.1038/s41598-019-52913-z), <https://doi.org/10.3389/fimmu.2019.01654>). Therefore, because of extrinsic apoptosis mediated tumor breakdown function of DR5 agonists, our data is

also supporting increased T-cell infiltration in tumors. However as stated above the higher stabilization and surface mobilization of PD-L1 in tumor microenvironment (after DR5 agonist treatment) limits T-cell function in these tumors despite enhanced infiltration. If we put 2 of these phenomena together, the concept is clear that targeting ICPs (such as PD-L1) by inhibiting immunosuppressive microenvironment will increase activity of incoming T-cells in these DR5 treated tumors. Similarly, since ROCK1 inhibitors indirectly reduce PD-L1 surface level which again reduces the immune suppression in tumor. However, based on our data we consistently see that DR5 agonist plus anti-PD-L1 targeting gave higher IFN- γ expression (Fig 6L, M), and more importantly higher survival (Fig 5Q). We believe the difference is due to different working mechanisms as anti-PD-L1 blocks both basal and overall PD-L1 function, while ROCK1i inhibitors only regulate intra-cellularly stabilized PD-L1 levels but does not reduce the basal PD-L1 levels in tumor microenvironment. All these conceptual points have also been updated in the introduction, results and discussion.

3. In fig1a the westerns for PD-L1 are not terribly convincing - quantification of replicates should be included. A PARP cleavage western should also be included in fig 1M. Where are the error bars and stats for fig 1E and K? Presumably the authors performed the experiment more than once? And how are there stats for 1G, but no error bars?

Response: We agree with the Reviewer. Yes, as per reviewer request, many western blots in Fig 1, Fig 2 etc have been replaced. Western blotting quantitation has been added for two different cell lines (n=3), please see Fig 1E and Appendix Fig S2A. We would also like to point out that rather than doing same experiments with same antibody 3 times in the same cell line, we have comprehensively repeated experiments in multiple tumor cell lines and have made use of multiple clinical DR5 agonists to support our hypothesis. Additional new data from two different cell line is present in Fig 1C and Fig EV1A. Similarly, for CSN5 independent PD-L1 stabilization western blots, additional data from another cancer line has been added in Fig 2G and Fig EV2F. In addition, the PARP cleavage blot has been included in Fig 1M (which is Fig 1K in the revised manuscript) as per reviewer's request. Error bar and stats have been updated in PD-L1 FACs data from cells and tumors which now are in Fig 1H, Fig 1J. in revised manuscript. PD-L1 IHC data has been also added in Fig 1L, Appendix Fig S2C. Importantly, statistical analyses and details have been added in all the figures wherever possible.

4. There is a huge variation in drug concentrations used in the T cell co-culture assays (e.g. from 5nM to 1uM for KMTR2). How many off-target effects may be driven by 200-fold increases in drug concentration? Can caspase inhibition still rescue T cell activation when using 1uM concentrations? Also, the authors can't say that these experiments confirm PD1-PD-L1 interaction. They should include PD-1 blocking antibody and determine whether the DR5 agonists no longer drive loss of T cell activity.

Response: We respectfully agree with the Reviewer concern that concentration variations have been used for some cell lines. The simple rationale for differential use of DR5 agonist concentration against different cell lines is their differential sensitivity to indicated DR5 agonist to

activate differential levels of caspase-8. For example: In our hands, OVCAR-3, MDA-MB-436, and Colo-205 are highly sensitive to DR5 agonist antibodies followed by U87 and HCT116 cells. A549 and MDA-MB231 cells are only partly sensitive to DR5 unless higher conc, which is why either 500nM or 1 μ M antibodies were used for these cells. Thus, in Fig 2C, D, the consistent 50nM concentration was used for OVCAR-3, MDA-MB-436 as they are similarly sensitive to DR5 agonists. To accommodate reviewer's concern experiments have been redone with a consistent 100nM concentration for colo-205 and U87 cells in Appendix Fig S4C. Similarly, experiments for MDA-MB-231 and A549 cells have been repeated with 500nM conc in Appendix Fig S4C.

Although Reviewer did not raise this concern, we would respectfully also like to point out that in some experiments instead of one antibody a combination of DR5 agonists were used. There are two reasons for that 1) DR5 agonist combinations have shown to increase their activity against partially sensitive cell lines. Please see- doi: 10.1016/j.ccr.2014.04.028, Cancer Cell 2014 Aug 11;26(2):177-89 . Therefore, some cell lines such as MDA-MB-231, A549 which are lesser sensitive to DR5 agonist, we activated DISC caspase-8 activity either by higher concentration or by using a combination of antibodies to support our hypothesis. 2) The second rational for these experiments was to show that PD-L1 stabilization is independent of particular antibody binding to particular epitope of DR5 rather because of higher DISC-caspase-8 activity. We also understand the rationale in the context can be confusing so we have removed redundant data generated from dual combinations wherever possible.

To answer reviewer's 2nd concern "the authors can't say that these experiments confirm PD1-PD-L1 interaction", we have done the experiments using syngeneic tumor studies and data is included in Fig EV4C-D and Fig 6L.

Since we are not aware of murine PD-1 cross-reactive clinical antibody, we made use of a PD-1 small molecular inhibitor (BMS202). In Fig EV4C-D, both lexatumumab + Avelumab and lexatumumab + BMS202 treatment significantly reduced tumor size as compared to lexatumumab alone (n=4=5 tumor bearing animals). Furthermore, since there was no statistical-significant difference between lexatumumab + Avelumab and lexatumumab + BMS202 treated tumors, these results confirm that reduced tumor size is because DR5 agonist can no longer drive the loss of T-cell function when added in combination of either anti-PD-L1 antibody or PD-1 inhibitor. In Fig 6L, IFN- γ expression (via flow cytometry) was similar in sized matched tumors treated either with MD5-1 + BMS202 and MD5-1 + ROCK1i.

In addition, the PD1+ stable jurkat cell, an immortalized human T-cell reporter assay (loss of luciferase activity) described in Fig 2A-C, directly confirms PD-L1-PD-1 interaction after co-culturing with DR5 agonist treated tumor cells.

5. In order to confirm that the EV transfer of PD-L1 is a DR5-mediated phenomenon, they should include non-DR5 targeting therapy here.

Response: We completely agree with Reviewer. It is another excellent suggestion by reviewer.

In response, since TNF- α treatment also stabilizes PD-L1, we isolated ApoEVs from TNF- α treated cells (MDA-MD-436 and OVCAR-3) alongside IgG1 and DR5 treated cells. After confirming optimal ApoEVs PD-L1 transfer kinetics (Fig 3G, Appendix Fig S6), ApoEVs from DR5, IgG1 and TNF- α treated cells were incubated with DR5-KO cell line. The data from two different cell lines is included in Fig EV3A, which support high PD-L1 transfer only from ApoEVs isolated after DR5 agonist treatment. ApoEV mediated PD-L1 transfer was similar between TNF- α and IgG1 treatment.

6. The authors state that ROCK1 cleavage was consistently observed prior to full caspase-3 activation. They cannot make that statement based on westerns of varying quality and intensity. They suggest that these data support a direct role of caspase-8 (and DISC) in ROCK1 cleavage, which they do not have evidence for. They should at least use a caspase 8 inhibitor to verify this. They also conclude that ROCK1 activation is a key regulatory second step in order to surface mobilize the internally stabilized PD-L1 on cell-surface - unless they conduct some live cell imaging experiments, they can't conclude anything about ROCK1's involvement in mobilization.

Response: We appreciate Reviewer's comment. Although our data from multiple cell lines (Fig 4, Fig EV3C-G) is highly consistent, we also agree with the Reviewer that a caspase-8 selective inhibitor would have strengthened the data. Unfortunately, there is no caspase-8 only selective inhibitor available; many commercial inhibitors that we have tested also inhibits caspase-3 despite stating that they prefer caspase-8. Thus, considering this lack of data and to accommodate Reviewer's concern we have toned down our language in the text to conclude caspase-8 being the potentially additional upstream caspase downstream of DR5 agonist, being involved in ROCK1 activation along with caspase-3.

Regarding, the statement "ROCK1 activation is a key regulatory second step in order to surface mobilize the internally stabilized PD-L1 on cell-surface" on Reviewer's request, we have removed "a key regulatory step" from the language although we highly respectfully partly disagree with the Reviewer analysis because of following reasons. Among these, the first 3 sets of data were provided in first submission:

a) As the data shown in Fig 4J, with multiple DR5 agonist antibodies, the luciferase activity in reporter assay is significantly lower when tumor cells were treated with multiple DR5 agonist antibodies. On the other hand, addition of DR5 agonist +ROCK1i significantly increased the luciferase activity (stats provided). Two different ROCK1 inhibitors were used and 3 different DR5 agonist were used. Since the assay in Fig 4J is highly dependent on surface presence of PD-L1, the results support ROCK1 inhibitor reducing PD-L1 surface levels.

b) Immunoprecipitation data in Fig 4G is also critical. As seen in Fig 4G (Right panel PD-L1 blot), treatment of anti-DR5 (KMTR2) vs anti-DR5 +ROCK1 inhibitor (GSK269962) both stabilized PD-L1 total levels (lane-4, 5) but its immunoprecipitation in pull down (Left panel PD-L1 blot) with Avelumab (anti-PD-L1) was much higher in anti-DR5 (KMTR2) vs anti-DR5 +ROCK1 inhibitor treated. Since avelumab binds and pull-down surface PD-L1, these results support lower surface presence of PD-L1 in anti-DR5 +ROCK1 inhibitor (GSK269962) treated cells as compared to anti-DR5 (KMTR2) treated. We also kindly request reviewer to look at the

middle panel that have leftover supernatant after IP (middle panel PD-L1 blot). PD-L1 is almost completely pulldown from anti-DR5 (KMTR2) treated lysates but not from anti-DR5 +ROCK1 inhibitor (GSK269962) treated lysates. These results further suggest decreased surface PD-L1 presence in anti-DR5 +ROCK1 inhibitor (GSK269962) treated samples despite being stabilized in the lysates. Similar results of higher PD-L1 pulldown were evident in another cell line when treated with DR5 agonist lexatumumab but were reduced when treated with anti-DR5 +ROCK1 inhibitor, GSK269962 (Middle PD-L1 blot, Fig 4H).

c) Data in Appendix Fig S7, where all 4 tested ROCK1i inhibitors reduced surface PD-L1 levels when treated in combination with DR5 agonist KMTR2. The data was from 2 different cell lines.

New data sets which indirectly support ROCK1 role in PD-L1 surface stability.

d) Similar to avelumab (which works by inhibiting surface PD-L1 activity), DR5 +ROCK1i inhibitor also improved granzyme-b expression (Fig 6D, EV4E) as compared to DR5 antibody alone treatment.

e) ROCK1i inhibition and PD-1 inhibition (BMS202) inhibition have same results in terms of IFN- γ expression on CD8 enriched T-cell (Fig 6L)

To further accommodate reviewer's request, we have added following in discussion:

"How ROCK1 help mobilize PD-L1 to cell surface needs further investigations. If activated ROCK1 makes use of same set of regulators that are important for its established membrane blebbing and cytoskeletal ruffling function (Coleman et al., 2001) or some other mechanisms that potentially also includes other regulator such as CMTM6 (Fig 4) to help mobilize PD-L1 to tumor cell surface is beyond the scope of these investigations CMTM6(Burr et al., 2017)".

7. When testing therapeutic strategies in vivo, the authors mostly show pictures of tumor size but don't suggest how long treatment was for - were these tumors excised at end point? At a specific timepoint?

Response: We agree with the Reviewer. Yes, specific treatment times, dose and when tumors were excised and other endpoints, all these details are now added to the manuscript. Please see the figure legends for Fig 5 and Fig 6.

8. In xenograft studies, the tumor size will affect TIL infiltration. The authors need to show PD-L1 expression, T cell penetrance and cleaved caspase 3 in vivo (and in situ e.g. by IHC, not from homogenized tumor), and preferably control for tumor size - either IHC to see where T cells are, or perform the T cell FACS in size matched tumors.

Response: We respectfully agree with the reviewer suggestion. In response significant new data has been provided from sized matched tumors, which supports increase immune T-cells infiltration in solid tumors by DR5 agonist.

Please see Fig 6A-C, Fig 6D, Fig 6K, Fig 6L-M, Fig EV5, Fig EV4E and Appendix Fig S10, Appendix S11A-C).

The data in Fig 6A-C and Fig 6L-M has been generated with n=6-20 tumor bearing animals and isolated size match tumors (3 Independent experiments).

The data in Fig 6D, Fig 6K and Fig EV4E supports high granzyme-b activity (an indicator of cytotoxic T-cell function) in tumors consistent with enriched CD8 signal.

The IHC Data in Fig EV5 also supports data in Fig 6A-C, Fig 6D, Fig 6K and Fig EV4E.

The Data in Fig 6L-M and Appendix Fig S11A-C support high IFN- γ expression (flow cytometry) from size matched tumors after various indicated treatments. These newly added data sets support DR5 agonist role in tumor breakdown to potentially increase TIL population in tumors.

9. The authors make the unusual claim that GSK269 reduced tumors growths only at high >10mg/kg dose, while 1mg/kg dose in their cellular assay was effective to inhibit PD-L1 mobilization. How exactly do they calculate mg/kg in a cellular assay?

Response: We agree with the reviewer that right choice of words was not used in that statement. Regardless “GSK269962 reduced tumors growths only at high >10mg/kg dose” data has been removed from the figure. The 10mg/kg dose data becomes irrelevant in the revised submission as unlike previous submission, multiple new data sets have been added now, all of which make use of GSK269962 inhibitor alone at 2mg/kg dose (Fig 5, Fig 6, Fig EV4). For every experiment, 2mg/kg dose of ROCK1 inhibitor was used in combination of DR5 agonists and whenever 2mg/kg dose of ROCK1 inhibitor alone was used we did not see any significant change in tumor size. The ROCK1 alone treatment with 2mg/kg dose has been done multiple experiments as shown in Fig 4K (n=4), Fig 5H, I (n=4), Fig 5L (n=4), Fig 5N, O (n=3), Fig 5P (n=5), Fig 6A, B, C (n=6).

Regarding dose calculation, we respectfully like to explain to the Reviewer. Considering 20 mg average weight of mice, we make a 1mg/ml solution of every drug and when injected 200 μ l in mice, it will be 10mg/kg and when injected 100 μ l of it will be 5mg/kg and when injected 40 μ l of it, it will 2mg/kg. For in vivo studies, we used 2mg/kg dose of ROCK1 inhibitor (40 μ l in PBS) in combination of DR5 agonist. We apologize for the wording, it should be “in vivo” assay instead of “cellular assays”. For cellular assays, standard 250 μ M dose was used. These dosing details has been made clear in the revised manuscript.

10. When examining the TILs, following treatment with the bi-specific antibody the authors claim that, as expected, MD5-1 treatment inhibited TIL mobilization into tumors. That is not what figure 6N shows. Can the authors explain this?

Response: We respectfully agree with the reviewer that we did not made the right choice of word in that statement. Regardless the Figure 6N has been removed and multiple additional new data sets have been added. Please see our response to point 8.

Minor points:

1. S2B axes don't make any sense. Neither does the legend: "(B) Same as B in colo-205 cells"
We totally agree with reviewer's suggestion. S2B axes and legend has been updated, which is now EV1D.

2. I wouldn't label DR5-sensitive cancer cells as 'WT'

We totally agree with reviewer's suggestion. "DR5-sensitive cancer cells" is now replaced with "WT" cells

3. In fig S3 the legend states 4-6 doses, figure suggests 2-3. Which is right?

We apologize for typo. 4-6 doses are the right legend.

4. Why is the ROCK western in fig S6A so different to the rest?

We apologize for the mislabeling. The blot in Fig S6A was PARP, which is now moved to Appendix Fig S4D in revised manuscript.

5. Fig S9 is labelled S10

We apologize for the mislabeling. Figure labelling has been updated as per EMBO MM requirements.

6. In the clinical agonist the authors fail to show PD-L1 expression in DR5 vs DR5/ROCK inhibitor treated tumors. I'm sure they have this data.

We appreciate reviewer's comment. Yes PD-L1 expression after DR5 vs DR5/ROCK1i treatment is shown in Figure 4G. This data was present during previous submission also.

Please see the right panel PD-L1 blot (Total before IP panel, lysates), second blot from the top. The last two lanes (lane-4 and lane-5) shows total PD-L1 after DR5 vs DR5/ROCK1i treatment

Dear Dr. Tushir-Singh,

Thank you for your assistance in clarifying the issue with figure aberrations in your manuscript. After a discussion with our Head of Scientific Publications, Bernd Pulverer, and Data Integrity Analyst, Erica Boxheimer, we evaluated your explanation of the source of image aberrations as satisfactory and concluded that they had no impact on the interpretation of the data. Therefore, I am pleased to inform you that we decided to proceed with peer-review of your manuscript pending following amendments:

- Please remove all duplicated/overlapping images from figures EV9 and EV10. Each sample should be presented with one representative image. Quantifications should be performed only if more than one sample is processed.
- In the figure legends, please specify exact number of replicates for each experiment.
- Please upload individual, high-resolution figure files and include the figure legends to the main manuscript. You can choose up to 5 EV figures, which should also be uploaded as separate files, with their legends added to the manuscript after the main figure legends. All other figures, with their legends, should be compiled in one "Appendix" file with a table of content. The figures should be renamed "Appendix Figure S1" etc. Please make sure to rename and call out all the figures in main manuscript text.
- Please include Tables S1-4 to the "Appendix" and rename "Appendix Table S1" etc.
- Please include "Material and Methods" section in main manuscript text.
- Please check "Author Guidelines" for more information about manuscript preparation.

You will receive a notification e-mail with a link to access your manuscript after it is placed back in the Author Approval Folder. After making the adjustments specified above, please resubmit your paper following the same steps as before.

Thank you again for your cooperation in this matter and I look forward to receiving your revised manuscript.

23rd Nov 2020

Dear Dr. Tushir-Singh,

Thank you for the submission of your revised manuscript to EMBO Molecular Medicine. I am pleased to inform you that we will be able to accept your manuscript pending the following final amendments:

- 1) Manuscript type: Please submit your manuscript as "Research article". The total character count and number of figures in your manuscript exceed the limitations for "Report" articles.
- 2) In the main manuscript file, please do the following:
 - Correct/answer the track changes suggested by our data editors by working from the attached/uploaded document.

***** Reviewer's comments *****

Referee #1 (Comments on Novelty/Model System for Author):

Identification of mechanisms responsible of possible clinical failure of DR5 agonist antibodies.
Potential clinical impact on combination therapy with blocking PD-1-PD-L1 axis
Many models to support the conclusion

Referee #1 (Remarks for Author):

The authors have satisfactorily addressed my various concerns

Referee #3 (Remarks for Author):

The authors have done their best to deal with all points, and most have been adequately addressed.

The authors performed the requested changes.

4th Dec 2020

Dear Dr. Tushir-Singh,

We are pleased to inform you that your manuscript is accepted for publication.

Corresponding Author Name: Jogender Tushir-Singh

Manuscript Number: EMM-2020-12716